# VISIR-1.b: ocean surface gravity waves and currents for energy efficient navigation

Gianandrea Mannarini and Lorenzo Carelli

CMCC, Centro Euro–Mediterraneo sui Cambiamenti Climatici, via Augusto Imperatore 16, 73100 Lecce, Italy

**Correspondence:** gianandrea.mannarini@cmcc.it

**Abstract.** The latest development of the ship routing model published in Geosci. Model Dev., 9, 1597-1625 (2016) is VISIR-1.b, which is presented here.

The new version of the model targets large ocean-going vessels by considering both ocean surface gravity waves and currents. To effectively analyse currents in a graph-search method, new equations are derived and validated versus an analytical benchmark.

A case study in the Atlantic Ocean is presented, focusing on a route from the Chesapeake Bay to the Mediterranean Sea and vice versa. Ocean analysis fields from data-assimilative models (for both ocean state and hydrodynamics) are used. The impact of waves and currents on transatlantic crossings is assessed through mapping of the spatial variability of the tracks, an analysis of their kinematics, and their impact on the Energy Efficiency Operational Indicator (EEOI) of the International Maritime Organization. Sailing with or against the main ocean current is distinguished. The seasonal dependence of the EEOI savings is evaluated, and greater savings during summer crossings with higher intra-monthly variability in winter are indicated in the case study. The total monthly-mean savings are between 2 and 12%, while the contribution of ocean currents is between 1 and 4%.

Several other ocean routes are also considered, providing a pan-Atlantic scenario assessment of the potential gains in energy efficiency from optimal tracks, linking them to regional meteo-oceanographic features.

## 1   Introduction

The strongest water flows are generally observed in ocean Western boundary currents, in tidal currents, in the circulation of straits and fjords, in inland waterways, and in the vicinity of river runoffs (Apel, 1987). Even in marginal seas and semi-enclosed basins rapid flows may develop along semi-permanent circulation features (Robinson et al., 1999). However, advances in operational oceanography have revealed a high level of variability of the water flow at numerous spatial and temporal scales (Pinardi et al., 2015). This is indicated by both ocean drifter data, which are also affected by wind (Maximenko et al., 2012), satellite altimetry, which just provides the geostrophic component of the currents (Pascual et al., 2006), and model computations, whose capacity to represent mesoscale variability depends on spatial discretisation among other factors (Fu and Smith, 1996; Sandery and Sakov, 2017). More recently, even animal-borne measurements have been used to characterise ocean

currents, particularly in the polar regions (Roquet et al., 2013). For these applications, capturing such a complexity is essential in contributing to the value chain of ocean data (She et al., 2016).

The impact of ocean currents on navigation can be examined from several perspectives.

One approach can be based on Ship drift (SD) and dead reckoning. Dead reckoning refers to the computation of a vessel's position by means of establishing its previously known position and advancing it, based on its estimated speed and course over elapsed time. In the study of Richardson (1997), SD was defined as the difference in the velocity vector between two position fixes and the velocity vector resulting from dead reckoning. In Meehl (1982) a similar definition of SD was given, with the specification that dead reckoning must be computed 24 h in advance of the latest position fix. Historically, SD represents the first method of mapping ocean currents.

In the contexts of robust control and dynamic positioning, currents and other environmental fields, such as gravity waves and winds, are regarded as a disturbance to be compensated for so an objective can be achieved, such as keeping the vessel's position and heading. To achieve this task, numerical schemes typically assume that such disturbance is constant in time (Fossen, 2012) or at least slowly varying with respect to the signal of interest related to the vessel's internal dynamics (Loria et al., 2000).

Path following, a specific problem of motion control involving steering a marine vessel or a fleet of vessels along a desired spatial path, can account for the presence of unknown, constant ocean currents in addition to parametric model uncertainty (Almeida et al., 2010). Constraints on path curvature or accelerations, e.g. in reference to the concept of "Dubins' vehicle" (Dubins, 1957), may also be considered in the path planning procedure (Techy et al., 2010), or in the control sequence (Fossen et al., 2015).

The impact of ocean currents significantly affects slow-speed vehicles, such as Autonomous Underwater Vehicles (AUVs) or Underwater Gliders. Zamuda and Sosa (2014) use Differential Evolution (DE), an evolutionary algorithm, for glider path planning in the area of Gran Canaria island. They demonstrate the superior performance of DE with respect to state-of-the-art genetic algorithms and compare the fitness of several variants of DE. Regional ocean model current have also been used in a stochastic path planner for minimising AUV collision risk (Pereira et al., 2013).

Bijlsma (2010), while showing to be sceptical about the quantitative impact of ocean currents on ship routing, has recently generalised his optimal control scheme, which was originally conceived solely for waves (Bijlsma, 1975), in order to include currents. However, no new numerical results are presented in Bijlsma (2010).

A reconstruction of the Kuroshio current by means of drifter data is used by Chang et al. (2013) to demonstrate that it can be exploited for time-gains when navigating between Taipei and Tokyo (about 1,100 nmi apart). Suggested deviations from the great circle (GC) track appear to be chosen ad hoc, without any automatic optimisation procedure. Nevertheless, the authors found that the proposed track, despite extra mileage, leads to time-savings in the 2 – 6% range for super-slow-steaming (12 kn) vessels. The largest savings are obtained for the South-West-bound track (against the Kuroshio).

Currents may also be exploited for optimising navigation between given endpoints with respect to various strategic objective (e.g. track duration , fuel oil consumption, or $CO_2$ emissions).

Lo and McCord (1995) report significant (up to 6 – 9%) fuel savings in the Gulf Stream (GS) proper region for routes with or against the main current direction. Routes of constant duration and constant speed through water were considered per

construction. The horizontal spacing of the current fields used varied from 5°down to 1/10°, with the best fuel consumption savings at the finest spatial resolution. Little detail on the solution method is provided, which appears to be a graph-search, while their computational domain is not affected by coastlines.

An exact method based on the level set equation was developed by Lolla et al. (2014) and it is able to deal with generic dynamic flows and not constant vehicle speeds through the flow. This is based on two-step differential equations governing the propagation of the reachability front (a Hamilton-Jacobi level-set equation) and the time-optimal path (a particle backtracking ordinary differential equation). The level set approach was extended to deal with energy minimisation by Subramani and Lermusiaux (2016) showing the potential of intentional speed reduction in a dynamic flow. This method appears to be quite promising, though it has not as yet been embedded into an operational service.

Other mathematical techniques are reviewed in the introduction of Mannarini et al. (2016a) and some will be mentioned in this manuscript's Sect. 3.1 to help verify the new numerical results.

In the latest edition of the World Meteorological Organization's guide to marine meteorological services, ocean and tidal currents are considered to be a key variable in the management of vessel fuel consumption (WMO-Secretariat, 2017).

The International Maritime Organization (IMO) recommends avoiding "rough seas and head currents" among the ten measures within the Ship Energy Efficiency Management Plan, or SEEMP (Buhaug et al., 2009). The SEEMP came into force in January 2013, and applies to all new ships of 400 gross tonnes and above. It is one of the main instruments for mitigating the contribution of maritime transportation to climate change (Bazari and Longva, 2011).

## 1.1 New contribution

The above review of the literature shows that the question of the impact of sea or ocean currents on navigation, despite its classical appearance, is still open. The results are difficult to compara because: *i)* they are not validated versus exact solutions; *ii)* with some exceptions, they do not declare the computational performance; *iii)* generally, their model source codes are not openly accessible; *iv)* they are limited to case study analyses on a specific date, without any assessment of seasonal and geographical variability in their quantitative conclusions; *v)* they generally cannot account for both surface gravity waves and ocean currents.

All these considerations have motivated the development of the VISIR ship routing model presented in this paper, which is organised into three main sections: The theoretical framework for inclusion of currents into the model is presented in Sect. 2; the verification of the numerics and computational performance is shown in Sect. 3; the case-studies, including an assessment of seasonal and geographical variability, are provided in Sect. 4. Finally, the concluding remarks in Sect. 5 are followed by the statement of the availability policy of the model source code and input datasets. In App. A the main incremental changes of VISIR-1.b are documented.

Throughout this manuscript "track" indicates a set of waypoints joining two given endpoints or harbours, in relation to departure on a given date, and the "route" or "crossing" indicate when there is no reference to a specific departure date. "Wave" is a short form of "surface gravity wave" and the shortcuts "*w*" for computations accounting for only waves and "*cw*" for both ocean currents and waves.

## 2   Method

This section comprises all theoretical and numerical advancements of VISIR-1.b, with respect to the previously published version (VISIR-1.a).

The basic hypotheses are described in Sect. 2.1. They result in the kinematic equations derived in Sect. 2.2. The equations are solved on a graph, and its navigational safety and resolution features are analysed in Sect. 2.3. Changes to the graph search method are given in Sect. 2.4. Finally, the vessel seakeeping and propulsion modeling, including an estimation of voyage energy efficiency, is reviewed in Sect. 2.5.

All model features that are not explicitly mentioned in this paper are unchanged from the previous version. A summary of the main changes to the VISIR-1.a code is provided in Tab. A1. New abbreviations and symbols are reported in Tab. 1 and Tab. 4.

### 2.1   Basic assumptions

VISIR optimisation corresponds to the top layer in a hierarchical ship motion control system. It determines long-term routing policies that affect the motion of the vessel, viewed as a particle. The related kinematics occur over long period of time in the lower control layer, corresponding to the motion control level, and determine the behaviour of the vessel as a rigid body under the influence of external forces and moments (cf. App. B).

In terms of the nomenclature used, "vehicle" is here used as a more general term than "vessel" for the theoretical results that do not refer to any specific ship feature. The term "flow velocity" is used for referring to the velocity resulting from either ocean surface current, tidal current, and nonlinear mass transport in surface gravity waves (Stoke's shift), or their composition. Also, when not otherwise specified, the qualification "over ground" is assumed for both speeds and courses.

### 2.1.1   Linear superposition

Assuming that a linear superposition principle holds for vehicle and horizontal flow velocity, the vector Speed Over Ground (SOG) of the vehicle is given by

$$\frac{d\mathbf{x}}{dt} = \mathbf{F} + \mathbf{w} \tag{1}$$

where $\mathbf{F}$ is the vehicle Speed Through Water (STW) and $\mathbf{w}$ the flow velocity. The symbol $\mathbf{F}$ reminds that such speed, due to energy loss in waves, is in general a function of both vehicle propulsion parameters and ocean state, cf. Mannarini et al. (2016a, Eq.21).

Eq. 1 is a "no-slippage" condition: the vehicle is advected with the flow. The rationale for this assumption is the experimental observation that ocean drifters (including vessels) very quickly adjust, i.e., in less than one minute, their speed to the flow (Breivik and Allen, 2008). At the present level of approximation, such adjustment is instantaneous (as no second derivatives of $\mathbf{x}$ appear in Eq. 1) and it is independent of vessel displacement (no vehicle mass in Eq. 1). In their optimal control methods Bijlsma (2010) and Techy (2011) make the assumption of linear superposition of speeds, as does Zamuda and Sosa (2014) as

a kinematic basis of an evolutionary approach for describing glider motion. In the context of vessel motion control, Fossen (2012, Eq.26) defines STW or relative speed as a linear composition of SOG and current velocity.

However, we note that the superposition principle in the form of Eq. 1 only refers to a surface flow and cannot accommodate a depth-dependent (horizontal) flow speed $\mathbf{w}(z)$. Thus, vessel speed relative to water should be calculated using the balance
between the overall drag by the fluid (Newman, 1977) and the thrust provided by the propulsion system. This can be significant for large draught vessels, especially those sailing in stratified waters (where the vertical profile of water velocity may exhibit both magnitude and direction changes, cf. Apel (1987)).

Finally, the aerodynamic drag on vessel superstructure is also neglected in Eq. 1.

### 2.1.2 Course assignment

Along the vessel path, course over ground (COG) may need to be constrained for navigational reasons (traffic constraints, fairways, shallow waters, or any other reason for preferring a specific passage), and in the computation of an optimal path, the algorithm (such as a graph-search method) may resort to spatial and directional discretisation, which again is a form of course assignment.

Making reference to Fig. 1, if COG has to be along $\hat{e}$, then the vehicle vector velocity must satisfy:

$$15 \quad \hat{o} \cdot \frac{d\mathbf{x}}{dt} = 0 \tag{2}$$

where $\hat{o}$ is a normal versor of $\hat{e}$.

To keep the course constrained as per Eq. 2, it is assumed that the shipmaster can act on the rudder(s) for modifying heading $\hat{h}$ until COG satisfies Eq. 2 and then report the rudder(s) to the midship.

### 2.2 Resulting kinematics

After defining the vector components of the water flow

$$\mathbf{w} = \quad ||\mathbf{w}|| \hat{w} = \quad (u, v)^T \tag{3}$$

and making reference to Fig. 1, the flow projections along ($\hat{e}$) and across ($\hat{o}$) vehicle course (in either polar or rectangular representation) respectively are:

$$w_{\parallel} = \quad ||\mathbf{w}|| \cos(\psi_e - \psi_w) = \quad u\sin(\psi_e) + v\cos(\psi_e) \tag{4a}$$
$$25 \quad w_{\perp} = \quad ||\mathbf{w}|| \sin(\psi_e - \psi_w) = \quad v\sin(\psi_e) - u\cos(\psi_e) \tag{4b}$$

where for both course $\psi_e$ and flow direction $\psi_w$ the nautical/oceanographic convention (i.e., "where-to" direction, clockwise from due North) is employed. Furthermore, the choice of orientation of the $\hat{o}$ axis in Fig. 1 implies that a current bears to port whenever $w_{\perp} > 0$.

Linear superposition Eq. 1 and the course assignment condition Eq. 2 result into two scalar equations that, upon definition of an angle of attack $\delta$ of the ship's hull through the water (cf. Richardson (1997)):

$$\delta = \psi_s - \psi_e \tag{5}$$

as the difference between the angle of vehicle heading ($\psi_s$ or HDG) and the COG, read

$$
\begin{align}
5 \quad S_g &= F\cos(\delta) + w_\parallel \tag{6a} \\
0 &= -F\sin(\delta) + w_\perp \tag{6b}
\end{align}
$$

with the unknown $S_g$ recognised as the vehicle SOG. Remarkably, Eq. 6a-6b could also be used to determine ocean current vector $\mathbf{w}$, given SOG, STW, course and heading.

As long as $F$ is non null, $\delta$ is given by

$$10 \quad \delta = \arcsin\left(\frac{w_\perp}{F}\right), \quad F \neq 0 \tag{7}$$

In presence of waves, $F$ is reduced due to the wave-added resistance and can be obtained from a thrust-balance equation as in Mannarini et al. (2016a, Eq.14). As $F$ is always nonnegative, Eq. 7 implies that $\mathrm{sgn}(\delta) = \mathrm{sgn}(w_\perp)$. In particular, in the case of a cross flow $w_\perp$ bearing to port, a clockwise change of vehicle heading is needed for keeping course, as in the example shown in Fig. 1.

15      Replacing $\delta$ into Eq. 6a, SOG is obtained:

$$S_g = w_\parallel + \sqrt{F^2 - w_\perp^2} \tag{8}$$

Eq. 8 shows that the cross flow $w_\perp$ always (i.e., independently of its orientation) reduces SOG, as part of vehicle momentum must be spent on compensating for the drift. The along edge flow $w_\parallel$ ("drag") may instead either increase or decrease SOG. Notice that the "cross" and "along" specifications refer to vessel course, differing from vessel heading by the (usually small) 20  amount given in Eq. 7. Also it should be noted that the condition

$$S_g \geq 0 \tag{9}$$

is not guaranteed in case of a strong counter-current. In a directed graph (as the one used in VISIR), a violation of Eq. 9 along a specific edge would imply that the edge is made unavailable for sailing along that direction.

An equation formally identical to Eq. 8 was retrieved by Cheung (2017) in the context of flight path prediction, with wind 25  replacing the ocean currents and plane true airspeed replacing vessel STW.

Furthermore, both Eq. 7 and Eq. 8 hold if and only if

$$|w_\perp| \leq F \tag{10}$$

If this is not the case, vehicle speed cannot compensate for ocean current drift. We note that Eq. 10 is satisfied even in case of a vehicle drifting along the streamlines of the flow field without any steering ($F = w_\perp = 0$). Eq. 1 then reduces to $d\mathbf{x}/dt = w_\parallel \hat{e}$,

and vehicle heading is aligned with COG, or:

$$\delta = 0, \quad F = 0 \tag{11}$$

Finally, by taking the module of both sides of Eq. 1 and approximating the l.h.s. with its finite difference quotient (thus leading to a first order truncation error), the graph edge weight $\delta t$ is computed as

$$5 \quad \delta t = \frac{\delta x}{S_g} \tag{12}$$

where $\delta x$ is the edge length. The weights $\delta t$ are then used for the computation of a time-dependent shortest path, using the same graph search method described in Mannarini et al. (2016a) and updated in this manuscript in Sect. 2.4.

## 2.3 Graph preparation

In this section we report the procedure for ensuring that the graph used by VISIR is safe for navigational purposes. A note on
use of non-regular meshes can be found in App. C.

Due to the non-convexity of the shoreline and the presence of islands, the maritime space domain is not simply connected. and thus not all graph edges correspond to navigable courses. To account for this, the following graph pruning methodology is used. It starts from the observation that in a large ocean domain most of the edges do not intersect the coastline. Thus, the procedure consists of the following three steps:

*i)* Retrieve the indexes of edges within a small bounding box around each coastline segment;

    *ii)* Check edges within the bounding box for intersection with the coastline;

    *iii)* Create all edges in the selected domain, pruning just those – from *ii)* – intersecting the coastline.

The *i)* step can be performed in a constant time with respect to the size of the maritime domain because the graph is based on a structured grid. Furthermore, it can use a lower-resolution version of the shoreline (cf. Sect. 4.1.2) while the *ii)* step must use
a higher-resolution.

Thus, when creating the graph, only the sea and land arcs that do not intersect the shoreline are included in the graph. When the code for track computation is then run, the next node on the graph is determined for each of the requested track endpoints (i.e., start and end location of the route), what is its next node on the graph. This can even be a land rather then a sea node. In the subsequent step, the graph arcs are screened for the condition $UKC = z - T > 0$ (Mannarini et al., 2016a, Eq.44). Thus,
if the start node was found on land ($UKC \leq 0$), no outgoing path from that node can be computed and VISIR quits with a warning. The coordinate of the requested endpoint must then be shifted by the VISIR user, so its next node is not on land any more. This requires improvements before it can be used operationally, but for the current assessment exercise, whereby the endpoints are chosen just once and then used for many computations (288 tracks per route, cf. Sect. 4.5), this approach is still acceptable.

In VISIR-1.a graph nodes were linked only to all other nodes that can be reached via either one or two hops. In this work, a larger number of hops $\nu$ is, however, allowed. This enables the angular resolution $\Delta\theta$ to be increased up to:

$$\Delta\theta = \arctan(1/\nu) \tag{13}$$

The $\nu$ value is also called the "order of connectivity" of the graph (Diestel, 2005). In Mannarini et al. (2018, in review) the point is made that the numerical solution of the shortest path problem on a graph converges to the numerical truth as $\nu$ is increased in roughly inverse proportion to graph mesh spacing $\Delta_g$[1].

The computational cost of VISIR-1.b graph generation procedure is linear in the total number of edges (from step *i*) of the procedure above) within all the bounding boxes around the shoreline. For a given number of nodes, the computation time for preparing a graph of order $\nu$ then scales as $\mathcal{O}(\nu^2)$. More information on the scaling of the method performance can be found in App. C.

## 2.4 Time interpolation of edge weights

As in VISIR-1.a, also in VISIR-1.b edge weights are computed out of Eq. 12.

The shortest path algorithm is still derived from Dijkstra's one, which is a deterministic and exact method (Bertsekas, 1998). The algorithm was made time-dependent under the assumption that no waiting times at the tail nodes are necessary, or the "FIFO hypothesis" (Orda and Rom, 1990). Furthermore, a new option is introduced in VISIR-1.b to conduct the time-interpolation of the edge weights. Here, the edge weights are not kept constant between consecutive time-steps of the input geophysical fields (currents and/or waves) but are estimated at the exact time the tail node is expanded by the shortest path algorithm.

In Mannarini et al. (2018, in review) it was shown that the effect of time-interpolation can be relevant wherever the environmental fields rapidly change between successive time steps. This is likely the case for daily averages of the wave fields (Sect. 4.1.4), which are used for the case studies (Sect. 4) of this manuscript.

Orda and Rom (1990) stated that, under the FIFO hypothesis, the worst-case estimate of the computational performance is, as for the static case, $\mathcal{O}(N^2)$, with the number $N$ of graph grid points considered[2]. However, Foschini et al. (2014) pointed out that in the presence of time dependent edge weights, the computational performance may degrade to become non-polynomial. The scaling of performance with time-interpolation is further investigated in Sect. 3.2 through a few empirical tests.

## 2.5 Vessel modeling

The VISIR-1.b vessel propulsion and seakeeping model is the same as in VISIR-1.a, but with a minor update. It is reviewed and updated in following Sect. 2.5.1-2.5.2. Furthermore, under the hypothesis of constant Engine Order Telegraph (EOT), an estimate of the voyage energy efficiency is provided in Sect. 2.5.3.

---

[1]We here refer to a regular latitude/longitude mesh with $\Delta_g$ spacing, distinguishing from its projection on planar coordinates, with a constant $\Delta_y$ spacing and a $\Delta_x$ depending on latitude.

[2]The performance could be improved to $\mathcal{O}(N \log N)$ in a codification making use of binary heaps (Bertsekas, 1998).

### 2.5.1 Vessel speed in a seaway

STW together with the ocean current velocity determines SOG (Eq. 1). SOG in turn determines the edge weights in the graph representation of the kinematical problem (Eq. 12). STW depends on the vessel propulsion system (MANDieselTurbo, 2011) and on the energy dissipated through hydrodynamic viscous forces, aerodynamic forces, ocean surface gravity waves and waves generated by the vessel through the water displacement (Richardson, 1997). However it is beyond the scope of this manuscript to develop a vessel propulsion and sea-keeping model more realistic than that in VISIR-1.a (Mannarini et al., 2016a).

That model considered the balance of thrust and resistance at the propeller, neglecting the propeller torque equation (Triantafyllou and Hover, 2003). In the resistance, a term related to calm water is distinguished from a wave-added resistance. The calm water term depends on a dimensionless drag coefficient $C_T$, which within VISIR should have a power-law dependence on vessel speed through water: $C_T = \gamma_q (\text{STW})^q$. For the wave added resistance, its directional and spectral dependence is neglected, and only the peak value of the radiation part is considered. The latter was obtained by Alexandersson (2009) as a function of the vessel's principal particulars, starting from a statistical reanalysis of simulations based on Gerritsma and Beukelman (1972)'s method. By only considering radiation and neglecting the diffraction term, wave added resistance may be underestimated for long vessels, with respect to the wavelength.

### 2.5.2 Vessel intact stability

In line with a IMO guidance (IMO, 2007), VISIR also uses sea-state information to conduct a few checks of a vessel's intact stability. In Mannarini et al. (2016a) an ongoing research activity into this topic was noted. Specifically, at that time the development of "second generation" stability criteria was proposed by Belenky et al. (2011). A recent Terms of Reference for updating the IMO stability Code (IMO, 2008) was published by the IMO Maritime Safety Committee (IMO, 2018c).

At present, VISIR includes checks of intact stability related to: parametric roll, pure loss of stability, and surfriding/ broaching-to at an intermediate level between IMO (2007) and the second generation criteria. Either intentional speed reduction (EOT<1, Tab. 1) or course change can be exploited by VISIR for fulfilling the stability checks (Mannarini et al., 2016a).

Following (Mannarini et al., 2016a, Sect.2.2.2 & pseudocode in App.A), all vessel speeds at any location and direction (i.e. on each of the $A$ edges) and any time ($N_t$ time steps) are computed ahead of path optimization. A time-dependent Dijkstra's algorithm (Mannarini et al., 2016a) can then manage all this spatially and temporally dependent information for computing the time-optimal paths. Its correctness is demonstrated by comparison with the path resulting from the benchmark solution in a dynamic flow field (Sect. 3.1.2, Fig. 2, Tab. 2). Similarly, edges that, for a given EOT, violate stability are pruned before the shortest path algorithm is run. Stability loss is assumed to be local in both space and time, no matter what the previous path is before the vessel sails through the edge violating stability. Thus, the edge is pruned only for that time step, ahead of path optimization.

Therefore in terms of vessel stability, the sole update in in VISIR-1.b is in the actual values of the vessel parameters and the parametric roll stability check. The new vessel parameters are suited for modelling a container ship and are listed in Tab. 4.

These values result in a STW dependance on significant wave height as in Fig. 4a and resistances as in Fig. 4b. For the parametric roll, the wave steepness criterion is generalised for vessels of $L_\mathrm{wl} > 100\,\mathrm{m}$ by implementing the piecewise linear function of $L_\mathrm{wl}$ given by Belenky et al. (2011, Eq.2.37). Thus Mannarini et al. (2016a, Eq.32) is replaced by

$$H_\mathrm{s}/L_\mathrm{wl} \geq \Sigma \tag{14}$$

where the critical ratio $\Sigma$ is given by

$$\Sigma = \begin{cases} 1/20 & \text{for} \quad L_\mathrm{wl} < 100\,\mathrm{m} \\ 1/3 \cdot (1/5 - L_\mathrm{wl}[\mathrm{m}]/2000) & \text{for} \quad 100\,\mathrm{m} \leq L_\mathrm{wl} < 300\,\mathrm{m} \\ 1/60 & \text{for} \quad L_\mathrm{wl} \geq 300\,\mathrm{m} \end{cases} \tag{15}$$

As the stability changes are maximized for a ship length close to wavelength (Belenky et al., 2011, Sect.2.3.3), the $\Sigma$ ratio also represents a critical wave steepness. Thus, Eq. 15 implies that it reduces at larger wavelengths, making the check on loss of stability in rough seas more severe than within the previous (VISIR-1.a) formulation.

### 2.5.3 Voyage energy efficiency

In this subsection the impact of track optimisation on voyage energy efficiency is estimated.

Following the Paris Agreement (UNFCCC, 2015), anthropogenic climate change is receiving increased attention at both International and regulatory levels. The Intergovernmental Panel on Climate Change recently published a special report on the greenhouse gases (GHG) emission reduction pathway, to limit global warming above pre-industrial levels to $1.5°$. It was noted that this would require rapid and far-reaching transitions in energy systems and transport infrastructure (IPCC, 2018).

The third IMO GHG study estimated the share of emissions from international shipping in 2012 to be some 2.2% of the total anthropogenic CO2 emissions (Smith et al., 2014). According to the EDGAR database, emissions from International shipping in 2015 were higher than the quota of two countries such as Italy and Spain put together (JRC and PBL, 2016).

In line with the United Nations sustainable development goal 13[3], an initial GHG reduction strategy was approved by the IMO in April 2018 (IMO, 2018b). It is layered into three levels of ambition, with the second one being "to reduce CO2 emissions per transport work, as an average across international shipping, by at least 40% by 2030, pursuing efforts towards 70% by 2050, compared to 2008". Implementation through short-, mid- and long-term measures is envisaged. The short-term measures include the development of suitable indicators of operational energy efficiency.

The IMO had previously introduced the Energy Efficiency Operational Indicator (EEOI) as the ratio of $CO_2$ emissions per unit of transport work (IMO, 2009b). There are several possible definitions of transport work, depending on vessel type. We have restricted our focus to a cargo vessel carrying solely containers, for which transport work is defined as deadweight (DWT) times sailed distance $L$. In order to estimate the quantity in the numerator of EEOI, the $CO_2$ emissions are taken to be proportional to fuel consumption (IMO, 2009b), ending with

$$\mathrm{EEOI} = \frac{C_F \cdot s \cdot P \cdot T}{DWT \cdot L} \tag{16}$$

---

[3]https://sustainabledevelopment.un.org/sdg13

where the $C_F$ is a conversion factor from fuel consumption to mass of $CO_2$ emitted, $s$ is the specific fuel consumption, $P$ is the engine brake power and $T$ the sailing time. Variations of $P$ are allowed by the VISIR algorithm, Sect. 2.5.2, while $s$ is assumed to be a constant.

If a track is plied at a constant $P$ (i.e., EOT=1), the emissions are then proportional to $T$ and the EEOI ratio $\rho_{\beta,\alpha}$ of two tracks between same endpoints and sailed with same DWT is given by

$$\rho_{\beta,\alpha} = \frac{\text{EEOI}_\beta}{\text{EEOI}_\alpha} = \frac{T_\beta}{L_\beta} / \frac{T_\alpha}{L_\alpha} \tag{17}$$

where the subscripts label the $\beta$ track being compared to the $\alpha$ track. Eq. 17 shows that $\rho_{\beta,\alpha}$ is the inverse ratio of the average speeds along the $\beta$ and $\alpha$ tracks. The EEOI relative change of $\beta$ to $\alpha$ track is then given by

$$\Delta(\text{EEOI})_{\beta,\alpha} = \frac{\text{EEOI}_\beta - \text{EEOI}_\alpha}{\text{EEOI}_\alpha} = \rho_{\beta,\alpha} - 1 \tag{18}$$

If the average speed in the $\beta$ track is higher than in the $\alpha$ track, then $-\Delta(\text{EEOI})_{\beta,\alpha} > 0$, i.e. a EEOI saving is achieved.

Depending on the subscripts $\alpha$ and $\beta$, different types of $-\Delta(\text{EEOI})_{\beta,\alpha}$ will be computed in Sect. 4.4.4 for analysing the benefit of the optimal tracks. A non-constant EOT is accounted for by VISIR. However, for the EOT=1 limiting case, the following general properties can be established:

    *i)* If vessel stability checks (Sect. 2.5.1) do not lead to any diversions, the mean speed along the optimal track is never lower than along the least-distance (or: geodetic ) track. Thus, related EEOI savings are always non negative, $-\Delta(\text{EEOI})_{\beta,\text{g}} \geq 0$;

    *ii)* Since currents can be either advantageous or detrimental to SOG (Eq. 8), savings of the optimal tracks of *cw*-type can have any sign with respect to optimal tracks of *w*-type, $-\Delta(\text{EEOI})_{\text{cw,w}} \lesseqgtr 0$.

Predicted and recorded EEOI for a trans-Pacific route are compared in Lu et al. (2015).

## 3    Verification and Performance

VISIR-1.b path kinematics described in Sect. 2 are used for the numerical computation of optimal paths on graphs. In this section, an assessment of VISIR-1.b numerics is provided by means of verification vs. analytical benchmarks (Sect. 3.1) and a test of its computational performance (Sect. 3.2).

### 3.1    Analytical benchmarks

For the verification, VISIR-1.b includes a verification option to run synthetic fields as the input, instead of those from data assimilative geophysical models (as described in Sect. 4.1), leading to analytically known least-time trajectories or "brachistochrones".

The remainder of the processing (generation of the graph, evaluation of the edge weights, computation of the shortest path) is identical for both synthetic and modelistic environmental fields. However, as identified in Sect. 3.1.1 and Sect. 3.1.2 below,

the synthetic fields are described in terms of linear coordinates. Thus, the spherical coordinates of the graph nodes are first linearised via an equi-rectangular projection.

### 3.1.1 Waves

The least-time route in presence of waves is computed using VISIR by assuming that waves affect the speed through water of the vessel, Sect. 2.5.1. For a static wave field, this leads to a STW that is not explicitly dependent on time. This allows for the least-time path problem to be formulated in terms of a variational problem.

Analytical solutions are available for a subclass of these problems, in which STW depends on only one of the spatial coordinates (Morin, 2007). In particular, if speed through water $F$ depends on the square root of the position, as in

$$F = \sqrt{2g(2\mathcal{R} - y)} \tag{19}$$

and the initial point is at $y = 2\mathcal{R}$, the least-time path is given by (an arc of) cycloid with $\mathcal{R}$ and $g$ parameters determining length and acceleration, respectively (Broer, 2014; Jameson and Vassberg, 2000). The cycloid presents a cuspid at the initial point as, because along a brachistochrone the region with $F = 0$ has to be quit first. The remainder of the path corresponds to refraction within layers of increasing speed or decreasing wave height, according to Snell's law.

The cycloidal benchmark was also exploited in Mannarini et al. (2016a), where the numerical error of VISIR-1.a in path shape and duration was ascribed to the limited angular resolution (a graph with $\nu = 2$ was used).

For VISIR-1.b, we compute graphs of higher connectivity (Sect. 2.3), allowing the cycloidal benchmark to be more closely to approached. The results are provided in Fig. 2a and Tab. 2. A relative error of less than 1 per mil in $T^*$ can be attained by only acting on graph connectivity. This improves on the accuracy of VISIR-1.a by about one order of magnitude.

The cycloidal solution exploits the fact that a functional of the spatial coordinate is minimised under some necessary conditions provided by the Euler-Lagrange equations (Vratanar and Saje, 1998). The hypotheses leading to these equations are not satisfied in the more general case where the integrand of the functional explicitly depends on time. Instead, an assessment of the VISIR solution in time-dependent waves was conducted by comparison with the numerical results of an exact method based on partial differential equations (Mannarini et al., 2018, in review). However, the verification of VISIR with time-dependent fields versus an analytical benchmark is possible in the absence of waves and the presence of currents, as described in the following Sect. 3.1.2.

### 3.1.2 Currents

The optimal control formalism provides the framework for computing extremals of a function, not only explicitly depending on spatial coordinates but also on time (Pontryagin et al., 1962; Bijlsma, 1975; Luenberger, 1979). As that the optimal path is controlled by a group of variables, an additional relation ("adjoint equation") holds. A variant of this approach, the Bolza problem, was used for the computation of optimal transatlantic tracks with a time-dependent STW by Bijlsma (1975). Due to topological constraints, some regions of the ocean are unreachable, and the method involves guessing the initial vessel course, which may hinder the implementation in an automated system. Another variant is the approach of Perakis and Papadakis

(1989), which accounts for a delayed departure time and for passage through an intermediate location. However, its outcome is limited to finding only spatially local optimality conditions.

Several benchmark trajectories are provided by Techy (2011) based on Pontryagin's minimum principle (Luenberger, 1979), which use vehicle heading as a control variable. In particular, in the presence of currents, and for a constant speed $F$ relative to the flow (analogous to STW in the nautical case), an analytical relation between vehicle heading (which is the control variable) and vorticity of any (point-symmetric) current field is demonstrated. The field is given by:

$$\begin{cases} u = & \Gamma x - \Omega y \\ v = & \Omega x + \Gamma y \end{cases} \tag{20}$$

where both $\Gamma$ and $\Omega$ may depend on time. For the case study (Techy, 2011, Example 3), the start and end points are set at the side of one equilateral triangle, and the third vertex is at the flow origin ($x = y = 0$). Finally, the duration $T^*$ of the least-time path is retrieved through an iteration on the initial heading.

Fig. 2b displays the VISIR.b solution to problem Eq. 20 for a case where $\Gamma$ is a non null constant (divergent flow) and $\Omega$ (one half of the vertical vorticity) linearly changes in time as per parameters of Tab. 2. The resulting optimal path changes its curvature, swinging on both sides of the geodetic track, which is crossed at about one third of its length, cf. Techy (2011, Fig.12). The elongation of the swinging is quite small, with the optimal path differing from the geodetic by less than 1% in length. This poses a challenge to the numerical solver on the graph, as many and accurate course variations are required over a short distance. Thus, it is not surprising to find that the graph mesh spacing $\Delta_g$ is more critical for achieving convergence than the graph order of connectivity $\nu$. However, this only holds if a time-interpolation of edge weights (Sect. 2.4) is used. Otherwise, no significant improvements in $T^*$ can be achieved, cf. Tab. 2. With VISIR-1.b, a minimum error of about 1.3% in $T^*$ is obtained for the graphs used.

## 3.2 Computational performance

The computational performance (Sect. 3.2.1) and RAM allocation (Sect. 3.2.2) of the new VISIR model version is assessed here. The major changes in the source code with respect to the already published version (Mannarini et al., 2016a), are summarised in Tab. A1. All the computations for collecting the data of this section were run on an iMac (Processor: 3.5 GHz Intel Core i7; RAM: 32 GB 1600 MHz DDR3). The results are displayed in Fig. 3. Here, the number of degrees of freedom (DOF) of a VISIR job is given by the product $N_t A$ of the number $N_t$ of time steps (i.e., days) and the number $A$ of graph edges. $A$ in turn depends on the number of grid points $N$ comprised within the geographical region selected and on the order $\nu$ of the graph. Jobs with DOF varying over more than four decades are considered, corresponding to graph orders $\nu \in \{1, 9\}$.

### 3.2.1 CPU time

Fig. 3a displays both the cost of computing only the optimal track via the shortest path algorithm and the total job cost from its submission to the saving of the results (rendering excluded). Cases without and with time interpolation of the edge weights (Sect. 2.4) are distinguished (Sect. 2.4). The CPU time for the optimal track increases almost nearly linearly with the DOF. Below $10^7$

DOF, a minimum delay of about $1\,\mathrm{min}$ can be noticed in the total job cost, which is due to I/O operations. All fitted parameters are reported in Tab. 3. Asymptotically, it is found that VISIR time-dependent optimal path algorithm (with time-interpolation active) can be run at a cost of less than $3\mu s$/DOF. For comparisons to other ship routing models, see App. D.

In any two-dimensional regular mesh, the number $N$ of graph grid nodes scales quadratically with the inverse mesh resolu-
5 tion, $N \sim (1/\Delta_g)^2$. For the series of experiments in Fig. 3, we varied $\nu$ as $1/\Delta_g$. When taken together, these two effects result into:

$$\mathrm{DOF} = A \cdot N_t \sim \nu^2 N \sim (1/\Delta_g)^4 = \mathcal{O}(N^2) \tag{21}$$

Thus, the empirically retrieved linearity of CPU time with DOF corresponds to a quadratic dependence in $N$. This is in fact the expected worst-case performance of a Dijkstra's algorithm (Bertsekas, 1998). In the presence of binary heaps, such an estimate
can be reduced to $N \log N$. This will be considered in future VISIR versions.

Without time-interpolation, the optimal path algorithm is about eight times faster, Fig. 3c. Furthermore, in the same panel the computational overhead from the use of currents besides waves is assessed. There is no overhead for the shortest path computations (red circles), as they use a set of edge weights of the same size for both cases in the inputs. Instead, edge weight values are determined through the specific environmental fields used (waves alone or also currents). Thus, the preparation of
15 the denominator in Eq. 12 causes an overhead for the total job (blue circles), which is up to 30% for the sampled DOF range. Starting from $\nu = 8$, a rise in the overhead is observed. To understand its origin, the RAM allocation is investigated in the following.

### 3.2.2 RAM allocation

Fig. 3b shows that peak RAM increases to about $3 \times 10^8$ DOF, where it saturates. Here, the computer's physical memory limit
is approached, which leads to swapping and to a degradation of performance, as already observed in Fig. 3c.

This is even more apparent in Fig. 3d, where the ratio of peak RAM for the *cw*- to *w*-type computations is displayed. Peak RAM allocation occurs – for large enough jobs – during edge weights preparation, prior to the run of the shortest path algorithm (cf. *ew* and *opt* phases in Fig. 3e.f). There is up to 50% extra RAM that needs to be allocated if ocean currents are considered. In fact, five environmental scalar fields must be considered (significant wave height, direction, and peak period; zonal and
25 meridional current), but the latter two are not used in the *w*-type computations. Thus, apart from noise being below $1 \times 10^8$ DOF, a drop of the *cw*-to-*w* peak RAM ratio is recorded, as the allocation for the *cw*-case saturates while, for the *w*-case, it is still significantly lower than such a limit and can grow further. Thus, from Fig. 3d it is possible to define a "computational efficiency region", for VISIR jobs with DOF lower than the one leading to the the drop observed in Fig. 3d. The computations in subsequent Sect. 4 are performed on a cluster with a RAM of 64 GB, which can operate in its efficiency region even for
larger DOF values.

To further clarify the memory space requirements of VISIR, we focused on its shortest path algorithm and collected and analyzed additional datasets, as described below. These consist of:

$d_1$) time series of RAM allocation of the VISIR Matlab job[4]

$d_2$) stopwatch timer readings at specific VISIR processing phases[5]

The $d_2$) dataset is then temporally offset by matching the end of the $d_1$) dataset. Finally, the resulting $d_2$) data are smoothed by thinning, which results in the plots displayed in Fig. 3.e-f below.

For each graph angular resolution (indexed by $\nu$ parameter) the timeseries exhibit different relative importance (both in terms of duration and RAM allocation) of the various processing phases. However, the $d_1$) and $d_2$) datasets confirm that, for $6 \leq \nu \leq 9$, the peak RAM is allocated during the edge weight computation (*ew* phase). Furthermore, the shortest path algorithm is run twice: in its static version (Dijkstra, 1959) for the computation of the geodetic track, and in a time-dependent version for the optimal track (Mannarini et al., 2016a). The latter requires the edge delays at $N_t$ time steps in the input , and this justifies the uphill RAM step between these two phases. Finally, Fig. 3.e-f proves that time interpolation does not affect RAM allocation but solely CPU time.

## 4    Case studies

In this section, the capacity of VISIR-1.b to deal with both dynamic flows and sea state fields in realistic settings is demonstrated using the ocean current and wave analysis fields from data-assimilative ocean models.

This section presents the environmental fields used for the computations, Sect. 4.1; a documentation of the principal VISIR model settings employed, Sect. 4.2; a description of the results on individual tracks of a given departure date, Sect. 4.3, the analysis of their seasonal variability within a calendar year, Sect. 4.4, and the extension of such analysis to several routes in the Atlantic Ocean, Sect. 4.5.

### 4.1    Environmental fields

VISIR-1.b uses both static and dynamic environmental fields obtained from official European and US providers. The static environmental datasets are of the bathymetry and shoreline. The dynamic datasets are of the waves and ocean currents. The specific fields used are described in the following subsections.

### 4.1.1    Bathymetry

The GEBCO 2014 bathymetric database[6] (Weatherall et al., 2015) is used in VISIR-1.b. Its spatial resolution is 30 $\mathrm{arcsec}$ or 0.5 $\mathrm{nmi}$ in the meridional direction.

---

[4]Using the shell command: `top | grep MATLAB >> RAM-timeseries.txt`
[5]Using the Matlab commands: `tic, toc`
[6]https://www.gebco.net/data_and_products/gridded_bathymetry_data/

### 4.1.2 Shoreline

The Global Self-consistent, Hierarchical, High-resolution Geography Database (GSHHG[7]) of NOAA (Wessel and Smith, 1996) is used in VISIR-1.b. There are 5 versions (c, l, i, h, f) of the database, with a resolution of about 200m in the best case. Depending on the geographic domain, VISIR-1.b uses different versions of the GSHHG for the generation the graph (Sect. 2.3).
This limits the generation time in the case of jagged coastlines, such as in archipelagic domains.

### 4.1.3 Wind

Meteorological fields have not as yet been used for computing VISIR-1.b tracks. Surface wind fields have only been used in VISIR-1.a for sailboats (Mannarini et al., 2015). Wind also directly affects also motor vessels through an added aerodynamic resistance and a heeling moment, which are mainly significant for vessels with a large superstructure, such as passenger ships
(Fujiwara et al., 2006). This will be considered in future VISIR developments. We have only used a NOAA- Ocean Prediction Center review of marine weather[8] for describing the synoptic situation affecting the ocean state during the periods of the case study of Sect. 4.3. An archive of surface analysis maps[9] is also considered.

### 4.1.4 Waves

Wave analyses are obtained through CMEMS[10] from the operational global ocean analysis and forecast system of Météo-
France, based on the third-generation wave model MFWAM (Aouf and Lefevre, 2013).

This uses the optimal interpolation of significant wave height from Jason 2 & 3, Saral and Cryosat-2 altimeters. The model also takes into account the effect of currents on waves (Komen et al., 1996; Clementi et al., 2017). Thus, surface currents from corresponding CMEMS product (see Sect. 4.1.5) are employed and used to force daily the wave model daily. The currents modulate wave energy and also cause a refraction of the waves propagation. The wave spectrum is discretized into 24 directions
and 30 frequencies in the $[0.035 - 0.58]$ Hz range. Classically, this is the realm of ocean surface gravity waves (Munk, 1951). The vessel intact stability constraints used in VISIR (Sect. 2.5.2) set a time scale given by the vessel natural roll period (usually up to about 20 s, or more than 0.05 Hz).

The spatial resolution is $1/12°$(i.e. 5 nmi in the meridional direction). Three-hourly instantaneous fields of integrated wave parameters from the total spectrum (spectral significant wave height, mean wave direction and wave period at the spectral peak)
are averaged in a preprocessing stage based on "cdo dayavg"[11] into daily fields. Neither Stoke's drift nor the partitions (wind wave, primary swell wave and secondary swell wave) are as yet used in VISIR. Due to a much larger fetch, the impact of swell is estimated to be more significant in the Southern than in the Northern Atlantic Ocean (Hinwood et al., 1982).

---

[7]https://www.ngdc.noaa.gov/mgg/shorelines/

[8]http://www.vos.noaa.gov/mwl.shtml

[9]http://www.wetterzentrale.de

[10]http://marine.copernicus.eu/

[11]https://code.mpimet.mpg.de/projects/cdo/

The wave dataset name is GLOBAL_ANALYSIS_FORECAST_WAV_001_027[12], and the product validation is provided by a companion document[13]. The datasets were downloaded from CMEMS at least 14 days after their date of validity, ensuring that the best analyses are used.

### 4.1.5 Currents

Ocean currents are obtained through CMEMS from the operational Mercator global ocean analysis and forecast system, based on the NEMO v3.1 ocean model, (Madec, 2008).

This uses the SAM2 (SEEK Kernel) scheme for assimilating, among others: Sea Level Anomaly, Sea Surface Temperature, and Mean Dynamic Topography (CNES-CLS13), among others. The spatial resolution is $1/12°$(i.e. 5 nmi in meridional direction). Daily analyses of surface fields are used in VISIR-1.b.

The dataset name is GLOBAL_ANALYSIS_FORECAST_PHY_001_024[14] and the product validation is provided by a companion document[15]. The datasets were downloaded from CMEMS at least 14 days after their date of validity, ensuring that the best analyses are used.

### 4.2 VISIR settings

For the results shown in this section, optimal tracks are computed on a graph with the order of connectivity of $\nu = 8$ (cf. Sect. 2.3) and mesh spacing $\Delta_g = 1/8°$. This graph resolution parameters are chosen to strike a compromise between track accuracy (i.e. spatial and angular resolution) and computational cost of the numerical jobs (see the discussion in Sect. 3.2). The computations refer to a container ship, and the parameters are reported in Tab. 4. The resulting vessel's performance in waves is summarised in Fig. 4.

### 4.3 Individual tracks

We first consider a transatlantic crossing in the northern Atlantic Ocean, between Norfolk, at the mouth of the Chesapeake Bay (37°02.5' N, 76°04.2' W) and Algeciras, just past Gibraltar Strait (36°07.6' N, 5°24.9' W). Both East- and Westbound tracks are considered, Fig. 5.

First of all, we note that the geodetic (or least distance) track is northwards bent, as it is to be expected from an arc of GC of the Northern hemisphere on an equi-rectangular projection. The track is piecewise linear and its Northern edge is flattened due to the finite angular resolution of the graph: $\Delta\theta \approx 7.1°$ from Eq. 13. However, as Tab. 5 reports, the error in the length of the geodetic route made by VISIR is only a few permil. This is comparable to the accuracy of the function for the computation of distances on the sphere (used in VISIR) compared to the ellipsoidal datum (which is more accurate, but slower).

---

[12]http://cmems-resources.cls.fr/documents/PUM/CMEMS-GLO-PUM-001-027.pdf
[13]http://cmems-resources.cls.fr/documents/QUID/CMEMS-GLO-QUID-001-027.pdf
[14]http://cmems-resources.cls.fr/documents/PUM/CMEMS-GLO-PUM-001-024.pdf
[15]http://cmems-resources.cls.fr/documents/QUID/CMEMS-GLO-QUID-001-024.pdf

For these tracks, meteo-marine conditions are first introduced, Sect. 4.3.1, and track spatial and dynamical features are then discussed in Sect. 4.3.2, along with the impact on vessel stability in Sect. 4.3.3, and their base metrics in Sect. 4.3.4.

### 4.3.1 Meteo-marine conditions

The synoptic situation in the northern Atlantic during the week following June 21st, 2017 (departure date for the Eastbound track) was dominated by the Azores High blocking descent of subpolar Lows to the middle latitudes. This led to relatively calm ocean conditions (significant wave height $H_\mathrm{s} < 5\,\mathrm{m}$) for most of the region involved in the Norfolk-Algeciras crossing.

In the week following February 16th, 2017 (departure date for the Westbound track) a Low with storm-force winds formed near ($41^o$N, $52^o$W) was observed, which then moved N, influencing wave direction on the 19th and 20th. On February 22nd another storm with waves of $H_\mathrm{s} > 8\,\mathrm{m}$ developed at ($37^o$N, $58^o$W).

In terms of the currents are concerned, we note that the Eastern edge of the crossing is N of Cape Hatteras and, thus, N of the GS branch known as the Florida Current (Tomczak and Godfrey, 1994).

### 4.3.2 Track spatial and dynamical features

The topological and kinematical features of the optimal tracks of the case study are discussed in this subsection.

**Tracks topology**

Four different solutions for the optimal tracks of the USNFK-ESALG route are given in Fig. 5 (red lines).

For the Eastbound voyage, when only considering waves (*w*-type, Fig. 5a) the optimal track is quite close to the geodetic track. This is due to the absence of waves of relevant height along the path during the crossing (about eight days, cf. Tab. 5). Discontinuities are seen between significant wave height fields at consecutive time steps (vertical stripes separated by dashed lines). This is enhanced by the daily averaging of the original three-hourly fields, cf. Sect. 4.1.4.

When the optimal track is computed for the same departure date and direction but also considers ocean currents too (*cw*-type), the solution is significantly modified, Fig. 5b. A diversion S of the geodetic track is computed by VISIR-1.b. This is instrumental in exploiting advection by the GS through velocity composition (Eq. 8). Despite being longer in terms of sailed miles, this track is faster than the geodetic one, Tab. 5. A closer look at Fig. 5b reveals that the optimal track averages between the locations of opposite meanders of the first six oscillations of the GS proper, at 72–63°W. Subsequent meanders, which are prone to extrude filaments (and thus more stretched in the meridional direction), are followed increasingly closely by the optimal track.

On the Westbound voyage of *w*-type (Fig. 5c) the optimal track takes diversions to both S and N of the geodetic track. This longer path can be sailed at an higher SOG than the geodetic track, because it skips both the storm in the North-Eastern Atlantic at $\Delta t = 1 - 4\,\mathrm{days}$ since departure and the the storm developing at $\Delta t = 6\,\mathrm{days}$, at the latitude of the arrival harbour.

The optimal track for the same departure date and direction but *cw*-type (Fig. 5d) leads to yet another solution with respect to the *w*-type track. It sails N of the geodetic at all times. The speed loss due to the encounter with the storm at $\Delta t = 2 - 3\,\mathrm{days}$ is balanced by the speed gains due to a meander of the North Atlantic current encountered at $\Delta t \approx 4\,\mathrm{days}$ at 44 $^o$N and by the benefit of sailing slightly further away from the rough sea than the corresponding *w*-type track at $\Delta t = 5 - 6\,\mathrm{days}$.

**Tracks kinematics**

To gain a deeper insight into the results, in Fig. 6 a few kinematical variables are extracted along both the optimal and geodetic tracks, for both *cw*- and *w*-cases.

Starting from the Eastbound route, Fig. 6a, the SOG of the *cw* optimal track differs greatly from corresponding geodetic track. SOG gains by up to more than 4 kn are experienced in the first half of the path, due to the GS. During the final part of the navigation ($\Delta t \approx 6.5\,\text{days}$), a SOG > 22 kn peak appears shifted in both tracks. This is the signature of the Atlantic jet past Gibraltar, which is encountered about 5 hrs earlier along the optimal track (cf. below Fig. 6c). Instead, the SOG does not appreciably differ when *w*-type optimal and geodetic tracks are compared. This is consistent with the spatial pattern seen in Fig. 5a.

The geodetic Westbound track displays heavy oscillations in SOG with two deep local minima at $\Delta t \approx 3; 6\,\text{days}$ (Fig. 6b). These correspond to the two storms NE and SW of the track mentioned earlier. The SOG differs from that along the geodetic track just at $\Delta t \approx 1.5 - 3\,\text{days}$ along the optimal track of *w*-type, and this is due to its initial northbound diversion. Starting from $\Delta t = 4\,\text{days}$ both optimal tracks significantly differ from the geodetic track, with the *cw* one being confirmed as enabling the larger SOG in the second part of the crossing.

In Fig. 6c-d the ocean flow component $w_{\parallel}$ along vessel course (Eq. 4a) is displayed. This quantity, together with its normal counterpart $w_{\perp}$, determines, through Eq. 8, the value of SOG. The difference between the optimal and the geodetic tracks is noticeable for both East- and Westbound navigation. In Fig. 6c it can be seen that the algorithm manages to encounter a nearly always positive (i.e. along the course) $w_{\parallel}$, which even exceeds 4 kn at the end of the first day. It is apparent that the same $w_{\parallel}$ oscillations are retrieved in the SOG linechart of Fig. 6a for $\Delta t < 3\,\text{days}$ and at the $\Delta t \approx 6.5\,\text{days}$ peak. For Westbound navigation, $w_{\parallel}$ is mainly positive (apart from the initial impact of the Atlantic jet before Gibraltar is passed) along the optimal track and is mainly negative along the geodetic, which sails against the GS. At $\Delta t = 4\,\text{days}$ a NW-bound meander of the North-Atlantic current is encountered, with a positive drag of up to 1.5 kn.

Finally, the angle of attack $\delta$ needed for balancing the cross flow $w_{\perp}$ (Eq. 5) is displayed in Fig. 6e-f. The track-average of $\delta$ is nearly zero, its maximum value is of the order of $10°$, and its amplitude is larger wherever $|w_{\parallel}|$ is larger. The oscillations of $\delta$ with a larger elongation are a signature of the crossing of strong meanders, as seen in the first half of Fig. 6e, and at $\Delta t = 4\,\text{days}$ in Fig. 6f.

Per Eq. 5, $\delta$ comprises both vessel heading and course fluctuations. As shown in Fig. 5, the latter are not too strong as compared to those of the geodetic track. Thus, the question is if the heading fluctuations corresponding to the $\delta$ signals in Fig. 6e-f are compliant with vessel manoeuvrability. The maximum module of the Rate Of Turn (ROT) of HDG is found to be $2.9°/\text{min}$ for the East- and $1.5°/\text{min}$ for the Westbound track of *cw*-type. These values are comparable to the IMO prescribed accuracy of $1.0°/\text{min}$ for onboard ROT Indicators (IMO, 1983). Thus, heading fluctuations computed by VISIR-1.b for this route should be feasible with respect to manoeuvrability.

### 4.3.3 Safety of navigation

The stability constraints given in Sect. 2.5.2 were checked for. However, some of them did not result in any graph edge pruning during the actual transatlantic crossing of the vessel under consideration (cf. parameters in Tab. 4). In fact, pure loss of stability was not realised as the threshold condition on significant wave height of Mannarini et al. (2016a, Eq.36) was not reached. Surfriding/broaching-to was not activated due to the condition that the Froude Number was never larger than the critical one for the wave steepness encountered (Mannarini et al., 2016a, Eq.42–43). By using the generalisation discussed in Sect. 2.5.2, parametric roll could instead occur for the present vessel parameters and the North Atlantic wave climate.

In addition, on this specific route and these departure dates, the voluntary speed reduction (Sect. 2.5.2) was not found to be activated by the algorithm. This mean that the tracks are sailed at a constant $P$ and that the $CO_2$ emissions are linearly proportional to the sailing time $T^*$ (Sect. 2.5.3). Instead, for other routes in the Atlantic, this is not always the case, cf. Tab. 7.

Furthermore, all time-dependent edge weights along the optimal tracks fulfil the FIFO hypothesis (Sect. 2.4).

### 4.3.4 Track metrics

Two simple metrics for summarising the kinematics of a track are here proposed: the optimal track duration $T^*$ and the corresponding length $L$ (not a starred symbol, as this length is not the object of the optimisation). For the geodetic tracks, optimisation is instead performed on length $L^*$ and, unless safety constraints play a role in the actual optimal track, the corresponding duration $T$ is higher than $T^*$.

$L$ is sensitive to the geometrical level of the track diversions, while $T^*$ reflects their kinematical impact. Such key metrics are reported in detail for both the geodetic and optimal tracks of both the East- and Westbound crossings in Tab. 5. The data also allow us to distinguish the quantitative role of waves and currents and the level of the track duration gains. For example, it is seen that both East- and Westbound tracks lead to time savings $\sim 3\%$ with respect to the geodetic track. However, for the former such a saving is mainly due to the exploitation of currents, while the latter is due to waves.

Concerning time gains, it is important to specify whether they refer to the geodetic track ($\Delta T_g$) or to an optimal track computed in presence of waves only ($\Delta T_w$). Here, we observe that both Lo and McCord (1995) and Chang et al. (2013), not using waves, only consider $\Delta T_g$. In addition, the model region chosen for their track optimisation almost coincides with the domain where the Western boundary current under consideration is at its strongest. This is different from the case study presented in this section, which also entails the Eastern part of the ocean, where the influence of the Western boundary current is less noticeable. Thus, the $\Delta T_g$ gains due to currents reported in Tab. 5 are lower than the results in the literature, although they are possibly more realistic because referring to full transatlantic crossings.

### 4.4 Track seasonal variability

In this subsection we consider to what extent the seasonal variability of the ocean state and circulation affects the variability of the optimal track of a given transatlantic crossing.

In order to address it, VISIR-1.b computations are conducted for departure dates spanning the whole calendar year 2017. Departures on six dates (1st, 6th, 11th, 16th, 21st, and 26th) in each month are considered, resulting in 72 dates per year. This is aimed at considering the decorrelation of the ocean current fields after a Lagrangian eddy timescale of about five days (Lumpkin et al., 2002). As waves are mainly driven by winds, whose velocity is one order of magnitude larger than ocean
velocities, the timescale for the decorrelation of the ocean state is expected to be even shorter.

To analyse the massive data resulting from these computations, four levels of analysis are considered: spatial variability of the tracks (Sect. 4.4.1), their kinematic variability (Sect. 4.4.2), the distribution of duration $T^*$ and length $L$, (Sect. 4.4.3), and the impact on voyage energy efficiency (EEOI, Sect. 4.4.4).

### 4.4.1   Spatial variability

A direct visualisation of the annual variability of the track topology is shown in Fig. 7.

Each panel displays a bundle of trajectories relative to the 72 departure dates. The extent of the diversions makes clear that the case study of Sect. 4.3 is not even extreme. Instead, for both East- and Westbound tracks, the summer and autumn tracks are closest to the GC track, because in the Northern Atlantic Ocean wave heights tend to be smaller in these seasons and, consequently, both vessel speed losses and relative kinematic benefits from diversions, are smaller.

Some tracks are found to sail quite inshore towards the Canadian coast, and for this we refer to a related comment in Sect. 4.5.4.

The general impact of ocean currents on Eastbound tracks is that the bundle of tracks squeezes and shifts S in the vicinity of the GS proper (W of 67°W). On a few dates (mainly in winter and spring) this is not the case, as storm systems happen to cross the location of the GS. For the Westbound tracks, accounting also for currents only adds a small perturbation to the wave-only
tracks, without dramatically changing their topology.

It should be stressed that the computed spatial variability depends heavily on how ship energy-loss in waves is parametrised, cf. Sect. 2.5.1. Wave-added resistance determines vessel STW for any given sea state, and thus how profitable a diversion to avoid speed loss is.

### 4.4.2   Evolution lines

While the paths of the tracks displayed in Fig. 7 convey the information about the spatial variability and its seasonal dependence, they fail to provide information about vessel kinematics along the tracks. Thus, an alternative visualisation is proposed in Fig. 8. Following a practice used in track anomaly detection (Zor and Kittler, 2017), cumulative sailed distance is displayed vs. time elapsed since departure. Thus, the slower parts of each path result in a smaller slope for corresponding segments of the track "evolution line". It can be seen that such slow segments are more frequent in winter months and in the middle of the
crossing, particularly for Westbound tracks, due to larger speed losses in waves.

Furthermore, in the presence of currents, the slope can exceed that relative to navigation at SOG equal to the maximum STW. This is due to the speed superposition per Eq. 8 and is apparent for some of the summer tracks in the panel relating to the Eastbound tracks in Fig. 8c.

Finally, the envelope of the evolution lines along the geodetic tracks is displayed as a grey etched area. This reveals the kinematical benefit of the optimal tracks, as they can be sailed at an higher SOG (coloured dots are generally left of the grey areas), resulting in shorter voyage durations.

### 4.4.3   Scatter plots

To reduce and better analyse the information contained in Fig. 8, the compound metrics $T^*$ and $L$ can be used, which are reported in a Cartesian plane in Fig. 9.

Such a plane contains a strictly forbidden region, left of $L = L_{\mathrm{GC}}$, which is the length (on the graph) of the GC arc connecting the route endpoints. The straight line through the origin, whose slope is $V_{\mathrm{max}}^{-1}$, generates another relevant partitioning of the plane. In fact, the region upper (lower) of this line corresponds to tracks sailed at an average speed lower (higher) than $V_{\mathrm{max}}$.

We first focus on Eastbound tracks. The distribution for $w$-type tracks is given in Fig. 9a. As expected, they are all comprised within the region above the $T^* = L/V_{\mathrm{max}}$ line. This is due to involuntary speed loss in a seaway, which reduces the average speed to less than $V_{\mathrm{max}}$. When currents are also considered, Fig. 9c, the tracks can be faster and, for Eastbound navigation, some of them even attain the region where the average SOG is larger than $V_{\mathrm{max}}$. This generally occurs for summer tracks, which experience a lower speed loss in waves.

For the Westbound tracks, Fig. 9b-d, the general picture differs in terms of the following features: the region where the average vessel SOG is larger than $V_{\mathrm{max}}$ is never attained and the distribution in the $(L, T^*)$ plane roughly maintains its pattern among the $w$- and $cw$-type results.

These findings are also mirrored in the Pearson's correlation coefficient $R_P$ between $T^*$ and $L$. While for the Westbound tracks $R_P$ is nearly unchanged (Fig. 9b-d), it decreases substantially between Fig. 9a-c. Most Eastbound tracks, independently of their duration, require a significant diversion to exploit the GS proper. This in turn reduces the correlation between $T^*$ and $L$.

The dots relative to the tracks selected for the featured analysis of Sect. 4.3 are circled in Fig. 9. For the Eastbound crossing, a transition into the efficiency region is seen when comparing the $w$-to the $cw$-tracks.

### 4.4.4   EEOI savings

For assessing the benefit of track optimisation in terms of voyage energy efficiency, in Fig. 10 the monthly and annual variability of the EEOI savings are displayed.

In reference to Sect. 2.5.3, specific fuel consumption $s$ is taken to be a constant, while engine brake power $P$ is allowed to vary as EOT is selected by the optimal routing algorithm (cf. Sect. 2.5.2).

With the notation of Eq. 18, EEOI savings of the tracks considering both ocean currents and waves ($\beta =$cw) are computed with respect to either the geodetic track ($\alpha =$g, Fig. 10a-b) or the wave-optimal tracks ($\alpha =$w, Fig. 10c-d).

For the Eastbound route, $-\Delta(\mathrm{EEOI})_{\mathrm{cw,g}}$ exhibits a clear seasonal cycle, with a peak of the monthly-mean value in winter. However, the winter intra-monthly variability exceeds the amplitude of the seasonal cycle. For the Westbound route, these trends are still observed, but both the seasonal cycle and the intra-monthly variability are less regular.

Furthermore, in Fig. 10c-d the monthly-mean value of $-\Delta(\mathrm{EEOI})_{\mathrm{cw,w}}$ is found to be larger for the Eastbound route, as it can benefit from advection by the GS. Peak values of $-\Delta(\mathrm{EEOI})_{\mathrm{cw,w}}$ are found in summer months, when the ocean state is calmer and thus the relative contribution of currents is the prevalent one.

Thus, the magnitude and location of the GS is critical for voyage energy efficiency along this route in Summer. In this respect, Minobe et al. (2010) found from satellite altimetry data show that the seasonal cycle of the geostrophic component of the GS is weak both in terms of meridional position and near-surface velocity. The simulations of Kang et al. (2016) instead show a seasonal cycle of the mean kinetic energy of the GS proper, with a relative maximum during summer. Berline et al. (2006) analysed the GS latitudinal position at 75–50°W from model re-analyses, and found that inter-annual and seasonal variability dominates upstream and downstream of 65 °W, respectively.

## 4.5 Ocean-wide statistics

The degree of optimization of actually sailed ship tracks is an open research question. Weather ship routing systems are used both offshore and onboard for planning, but the final decision is up to the shipmaster (Fujii et al., 2017). Furthermore, route planning may involve sensitive commercial information that a ship operator will not easily share. Thus, the extent to which a ship track is optimized is not always publicly known. We recently addressed this question by comparing VISIR optimal tracks based on wave analysis fields vs. reported ship tracks per AIS (Automated Identification System) data, for a route in the Southern Ocean (Mannarini et al., 2019). By computing both spatial and temporal discrepancies between VISIR and AIS tracks, we could infer that optimization likely took place in several but not all tracks.

VISIR can be used with either analysis or forecast environmental fields, as it is not constrained by any of the equations of Sect. 2. Thus, VISIR can help both predict optimal tracks (as actually done in the operational system for the Mediterranean Sea described in Mannarini et al. (2016b)) or assess past tracks (as we do in the present work). Transatlantic crossings may in some cases be longer than ten days, and thus exceed the maximum lead time of wave forecast model outputs, which are limited by the availability of related atmospheric forcing fields. The lead time of CMEMS products is limited to ten days for ocean current forecasts and to just five days for wave forecasts (cf. Product User Manuals cited in Sect. 4.1). To our knowledge, although ECMWF[16] runs a global wave model based on WAM with a ten-day lead time, it has a lower spatial resolution (1/8°) and no open access policy, while NCEP[17] runs a model based on WW3 on various grids and a lead time of 7.5 days[18].

The unavailability of long enough forecasts can be addressed by either re-routing or using supplementary information. Re-routing or re-planning involves the dynamic updating of the optimal track as new information (forecast) is made available (Stentz et al., 1995; Likhachev et al., 2005). The corresponding solution is sub-optimal, as the initial routing choices are unrecoverable and may compromise the attainment of a global-optimal solution. An example of the use of supplementary information instead has been proposed by Aendekerk (2018). Here, a "blending" of climatologies and geometrical information is used as a surrogate for missing forecasts with long lead times.

---

[16]https://www.ecmwf.int/en/forecasts/datasets/set-ii
[17]https://polar.ncep.noaa.gov/waves/hindcasts/prod-multi_1.php
[18]https://www.ncdc.noaa.gov/data-access/model-data/model-datasets/global-forcast-system-gfs

In a non-operational mode, the unavailability of forecasts is not critical. Analysis fields can then be used, enabling a better reconstruction of the environmental state. A product derived from analyses may be quite useful for scenario assessment, but the uncertainty associated with forecasts (Bos, 2018) complicate its usefulness. Analysis fields of waves and ocean currents are used throughout the present manuscript.

For 9 ordered couples of harbours from the list in Tab. 6, 72 tracks relative to year 2017 are computed. Two sailing directions and both *w* and *cw* cases are considered, leading to the computation of 288 tracks/route/year. This results in the computation of more than 2,500 tracks in the Atlantic Ocean with the same VISIR-1.b code version.

This exercise demonstrated the generality of the VISIR-1.b code for assessing the potential EEOI savings depending on various wave and ocean circulation patterns. This required among others that graph, shoreline, bathymetry, and environmental

datasets of waves and ocean currents to be made available for wide enough regions of the Atlantic Ocean, to account for the spatial variability of the tracks.

By using Tab. 7 and Fig. 11, the obtained general results can be summarised as follows:

a) EEOI savings in the Northern Atlantic are dominated by waves, with a contribution from currents that is not negligible. At the Equator, currents are the main reason for EEOI saving. In the Southern Atlantic, the largest savings are computed,

and they are mainly due to waves;

b) Routes mainly affected by ocean currents exhibit a large reduction of the correlation coefficient $R_P$ when comparing *w*- to *cw*-type scatter plots of track duration vs. track length;

e) The FIFO hypothesis is not satisfied in just a tiny number of edges, which are not used for the optimal tracks. This supports the use of a time-dependent Dijkstra's algorithm, as in Sect. 2.4;

20     d) Intentional vessel speed reduction (EOT<1) occurs in just three routes and for a relatively limited proportion of their track waypoints. This supports the approximation conducted in Sect. 2.5.3 for estimating the relative EEOI savings;

e) Maximum ROT never exceeds $20°/\min$. Given that COG changes are smooth (cf. e.g. Fig. 5), ROT changes reflect the HDG adjustments for balancing either strong or variable cross currents.

Route-specific results are discussed in the following paragraphs. In the Supplementary Material of this manuscript related

figures are published, and the web application for interactive exploration is available at http://www.atlantos-visir.com/. The application allows for the zooming-in to optimal tracks, checking their capacity in landmass avoidance and obtaining the EEOI savings compared to the least-distance track.

### 4.5.1   Buenos Aires - Port Elizabeth

The geodetic track is bent southwards in the Mercator projection. The (Northern Hemisphere) winter tracks are closer to the

geodetic, while summer ones exhibit greater diversions. This route is characterised by the highest impact of waves on energy efficiency savings. This can be ascribed to the strength of the Antarctic circumpolar winds, causing large waves in the Southern

Ocean (Lu et al., 2017). The role of currents on EEOI savings is instead about 1%, with a stronger contribution from the Benguela current for Eastbound crossings. This is generally due to the avoidance of the Agulhas current past Cape Town.

### 4.5.2 Equator route

This route does not join any major harbour and is just meant for sampling the Equatorial currents. In fact, the *w*-type optimal tracks are quite close to being an arc of the Equator. Nearly all of the optimal Eastbound *cw*-type tracks instead divert up to 5°N. This is for skipping the North Equatorial Current and exploiting wherever possible the Equatorial Counter-Current. However, the Westbound tracks make use of the North Brazil Current, diverting either N or S of the Equator by up to 3°.

### 4.5.3 Norfolk - Algeciras

This is the route discussed in the featured case study of Sect. 4.3. As this confirms, the route is affected to an appreciable extent by both waves and currents. The Gulf Stream significantly increases the efficiency of the Eastbound crossings and a clear seasonality of the EEOI savings is observed.

### 4.5.4 New York City - Le Havre

At their Western edge, these optimal tracks tend to sail inshore of Nova Scotia and Newfoundland and in some cases even in the Gulf of Saint Lawrence (Canada), also experiencing the effect of the Labrador current. This solution may not be viable in practice for two reasons. First, in Winter, sea ice can extend several tens of miles off the coastline. Second, coastal Canada is part of the Emission Control Areas (ECAs, IMO (2009a)), which may induce vessels to sail normal to the shoreline to more quickly leave the ECA. Neither effects are presently modelled within VISIR.

### 4.5.5 Santos - Mindelo

This route spans across both Hemispheres. The optimal tracks of *w*-type do not significantly differ from the geodetic track, with the Equator crossed at about 31°W. However, as ocean currents are also accounted for (*cw*-type), the crossing occurs within the 33-29°W band, depending on the actual strength of the North Brazil Current.

### 4.5.6 Mindelo - Genoa

This route connects the Atlantic Ocean to the Mediterranean Sea. In both sailing directions, it is dominated by waves. The tracks of *cw*-type are influenced by both the Atlantic jet past Gibraltar and the Canary current. They approach the energy efficient region (Sect. 4.4.3), particularly at the end of summer and in autumn. Topologically, they can sail very close to the shores of Morocco and the Western Sahara.

### 4.5.7 Rotterdam - Algeciras

This route links the major harbour of the Atlantic (Tab. 6) to the Mediterranean. The optimal tracks only slightly divert from the geodetic one, sailing close to some of the major west-European capes (Gibraltar, Cabo da Roca, Finisterre, North-Western Brittany, and the Strait of Dover). On just one date (Feb 1st, 2017) the optimal track sails several tens of miles inshore into the Gulf of Biscay, whether ocean currents are accounted for or not. This is due to the activation of the parametric roll safety constraint (Sect. 2.5.2), as the encounter period of waves is about half the natural roll period $T_R$ of the vessel, Tab. 4. This occurs only for the track leaving from Rotterdam, as waves are encountered at a lower frequency on the other sailing direction.

### 4.5.8 Miami - Panama

The spatial variability of this route is dominated by currents, as waves from sub-polar Lows are not relevant in the Caribic region. The bundle shows a waist W of Cuba (21°52' N, 85°00' W), a point through which all optimal tracks but one sail. In fact, on Sept.11th, 2017 the track leaving Miami is affected by large waves in the Gulf of Mexico generated by the transit of Hurricane Irma[19]. Here, the sea state, together with a local intensification of the GS in the Florida straits, leads to an optimal track sailing E of Cuba.

### 4.5.9 Boston- Miami

This route is heavily influenced by the Florida current. The Northbound tracks tend to align with the ocean flow. The South-bound tracks (sailing against the main flow) split into two sub-bundles, W and E of the Florida current. The Western sub-bundle is populated by mainly winter tracks. In fact, these tracks sail more inshore, avoiding the rough ocean state and thus reducing the speed loss in waves.

## 5 Conclusions

The VISIR ship routing model and code have been updated to Version 1-b. Optimal tracks can now be computed in the presence of both time-dependent ocean currents and waves. Vessel interaction with currents is described in terms of new equations which are validated by means of an analytical benchmark. To represent vessel courses with a higher degree of accuracy, the previous model version has been improved with respect to the capacity of computing graphs of a higher order of connectivity, thus accounting for the shoreline. The computational cost and memory allocation of the new model version is also assessed, and the inclusion of ocean currents leads to a total CPU time overhead not exceeding 30% for realistic computations (Fig. 3c).

While the code of VISIR-1.a was tested through its operational implementation in the Mediterranean Sea (Mannarini et al., 2016b), the robustness of VISIR-1.b has been proven through the computation of more than 2,500 tracks via the same model code version, spanning nearly all subdomains of the Atlantic Ocean.

---

[19]https://en.wikipedia.org/wiki/2017_Atlantic_hurricane_season

Several routes are considered, and the variability of the optimal tracks is mapped across a full calendar year (2017). Both spatial and kinematical variability of the tracks are accounted for, through various types of diagrams. The optimal exploitation of ocean currents may in some cases lead to average speeds greater than the maximum vessel speed in calm water (cf. Fig. 8-9). Finally, a standard voyage efficiency indicator (EEOI, introduced by the International Maritime Organization) is used to highlight the contribution of ocean currents and waves to the efficiency of the voyages. In some cases, EEOI relative savings were in excess of 5% (annual averages) and 10% (monthly averages), cf. Fig. 10–11. However, the intra-monthly, seasonal, and regional dependence of these results is quite high, and this study provides one of the first attempts to quantify it. It should also be noted that these results depend on the actual parametrisation of wave-added resistance, which is still formally the same as those of Mannarini et al. (2016a). These quantitative assessment of EEOI savings through path optimization may be considered in terms of the ongoing discussion at IMO-level about comparing the effectiveness of several proposed methods for vessel emission savings (IMO, 2018a).

Furthermore, the analysis of the track dataset is simplified by means of metrics such as the optimal track duration and length, their Pearson's correlation coefficient, and the maximum rate of turn of vessel heading. The correlation coefficient carries a signature of ocean currents, which tend to make optimal track duration and its length less correlated to each other. Furthermore, the approximation of a FIFO network (Sect. 2.4) is monitored and found to be satisfied to a great extent (Tab. 7). Vessel EOT is allowed to vary (Sect. 2.5.2), and the computation of the EEOI savings do account for this. However, intentional speed reduction is found to be a rare choice of the optimization algorithm.

We regard the main computational limitation of VISIR-1.a,b to be its requirement on computer RAM allocation (Sect. 3.2.2). The code still requires the preparation of all the time-dependent graph edge weights, ahead of the shortest path computations. This presently affects the capacity to describe the environmental state surrounding the vessel. For example, in this work we averaged three-hourly wave fields to daily averages (Sect. 4.1.4), but neglected other wave spectrum components (such as swell), nor we did account for the Stokes's drift contribution to the flow advecting the vessel.

However, it should be noted that a more realistic representation of the marine state is likely to correspond to a more accurate description of the mechanical interaction between it and the vessel, particularly with reference to speed loss in waves and wind (Tsujimoto et al., 2013), (Bertram and Couser, 2014). The presence of sea ice and ECA zones may also affect the optimal tracks. While the former effect may decrease in significance due to global warming, the latter has the potential to shape increasingly more coastal traffic, as the new IMO global cap on sulphur contents enters into force (IMO, 2016). Developing the representation of some of these model components is planned for future VISIR versions (e.g. in the frame of the newly started GUTTA project[20]) and will pave the way for end-to-end model evaluation exercises with respect to actually sailed trajectories.

*Code and data availability.* VISIR-1.b is coded in Matlab 2016a, which was used on both the workstation (Mac OS 10.11.6 "El Capitan", used for the performance analysis of Sect. 3.2) and the cluster (Unix CentOS release 6.9 "Final", used for the mass production of Sect. 4).

---

[20]http://bit.ly/guttaproject

In addition, the MEXCDF library is required. The list of all third-party Matlab functions is provided along with the VISIR-1.b release. The source code of VISIR-1.b is released with a LGPL licence at https://doi.org/10.5281/zenodo.2563074.

The additional figures referred to in Sect. 4.5 are part of the Supplementary Material. Support data assets for the figures and tables of this manuscript can be found at https://doi.org/10.5281/zenodo.3258177

*Author contributions.* Adopting CRediT (https://casrai.org/credit/) taxonomy:

G.M.: Conceptualization, Methodology, Software, Supervision, Validation, Visualization, Writing - original draft, Writing - review & editing;

L.C. : Methodology (Sect. 2.3), Software, Visualization.

*Competing interests.* Links MT together with CMCC run the operational service www.visir-nav.com, for which both a free and a premium version exist: The authors declare no competing interests with it. Furthermore, the term "VISIR" is a trademark of CMCC registered at EUIPO, https://euipo.europa.eu/.

*Disclaimer.* Research results not to be used for navigation: Neither the authors nor CMCC are liable for any damage or loss to assets or persons deriving from use of tracks computed by VISIR.

*Acknowledgements.* The work has received partial funding from the European Union's Horizon 2020 research and innovation programme under grant agreement No 633211 (AtlantOS). Furthermore, we would like to think Nadia Pinardi (University of Bologna) for her advice on the validation of the model; Fabio Montagna (CMCC) for consultancy on graph indexing; and Florian Aendekerk (Compagnie Maritime Belge) for providing realistic parameters of a container ship.

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

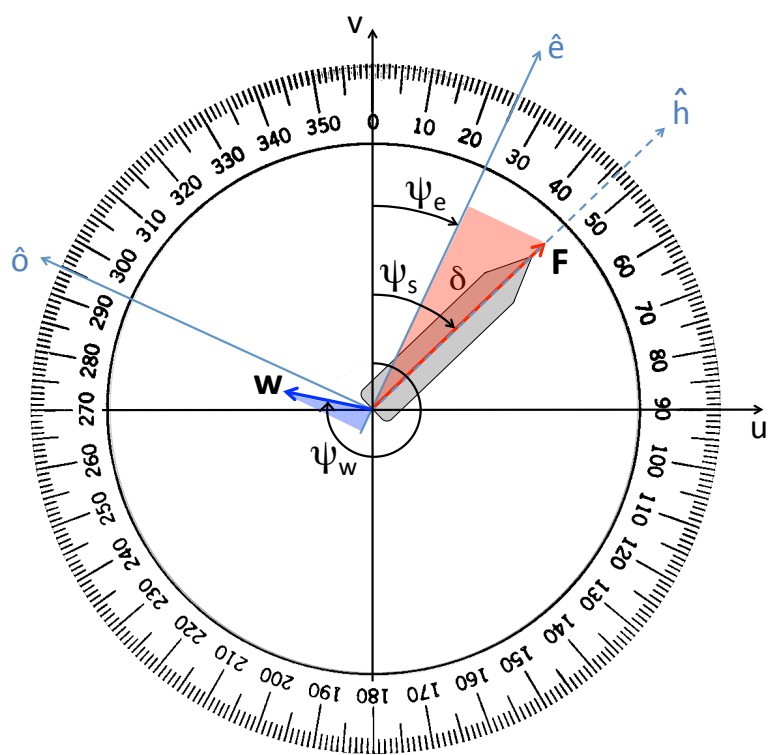

**Figure 1.** VISIR-1.b directional conventions on top of the compass protractor. Vessel speed through water $\mathbf{F}$, flow speed $\mathbf{w}$, vessel COG $\psi_e$, vessel heading $\psi_s$, flow direction $\psi_w$, angle of attack through water $\delta = \psi_s - \psi_e$. The longer (shorter) cathetus of the blue (red) triangle is the cross current magnitude $w_\perp = F \sin \delta$, cf. Eq. 7. The configuration displayed refers to a vessel course assignment $\psi_e = 25°$ and implies a positive angle of attack $\delta = 21°$ which balances the drift due to a port-bearing flow $\mathbf{w}$.

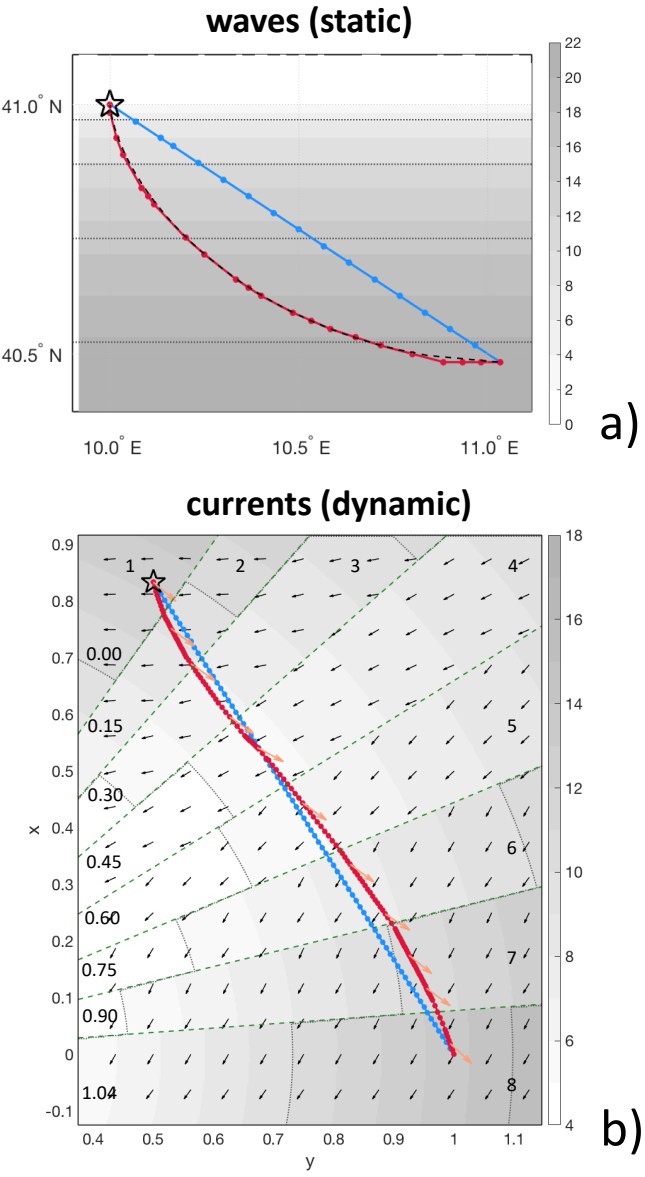

**Figure 2.** Verification of VISIR-1.b vs. benchmark solutions. Both least-distance (blue) and least-time (red) trajectories are displayed and the tracks originate at the black star symbols. a) A static wave field as in Eq. 19; the analytical solution (branch of an inverted descent cycloid) is portrayed as a dashed black line, cf. Mannarini et al. (2016a, Fig.9). b) A time-dependent current field as in Eq. 20; the vehicle heading is portrayed as orange arrows. The radial sectors separated by green dashed lines refer to a sequence of time-steps in the field, which are numbered in the outer sector. On the other hand, sector-mean times in unit of $T_0$ are given in the inner. Sectors # 3, 6, 8 should be compared to Techy (2011, Fig.12.a-c). In both a) and b) velocity field isolines every 5 kn are displayed as dots. Parameters of the synthetic fields are given in Tab. 2.

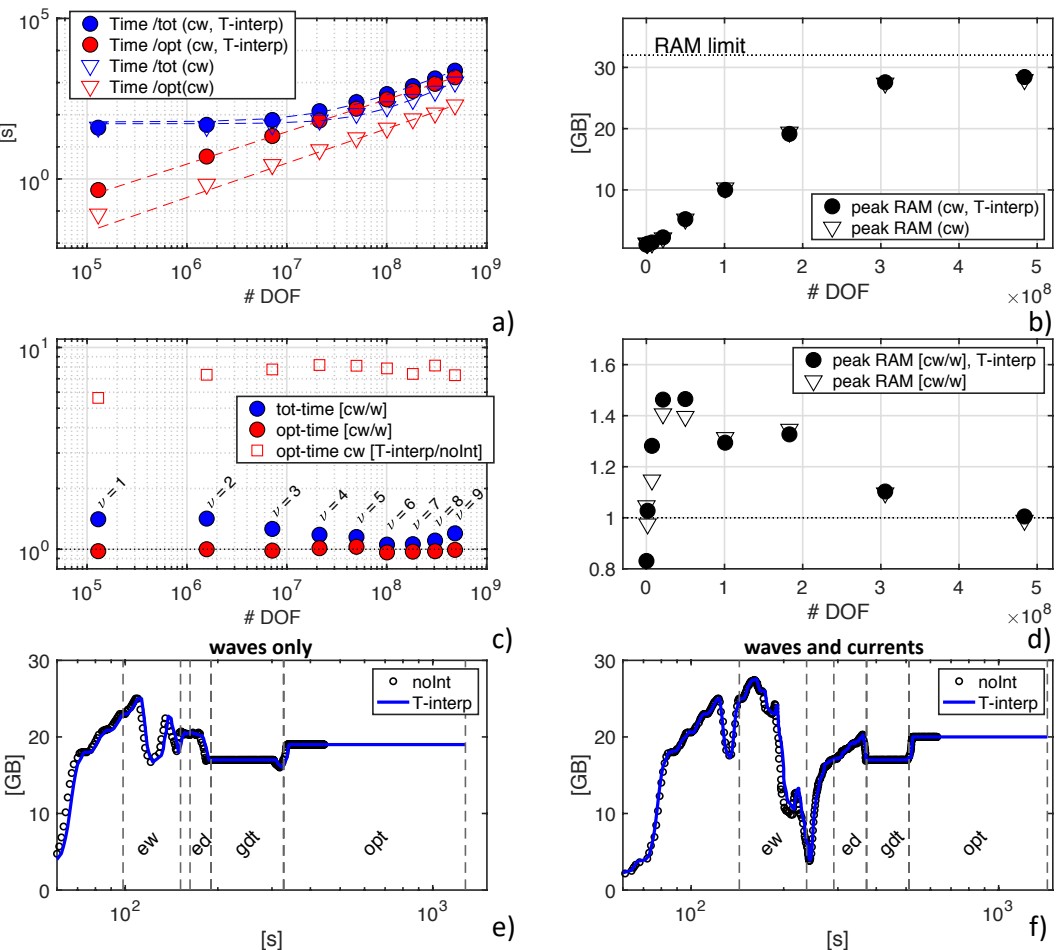

**Figure 3.** a) CPU time for the total VISIR job (blue markers) and for just the computation of the time-dependent shortest path (red markers). Only the *cw* case is considered. Dashed lines are fits of the model in Tab. 3. b) Peak RAM allocation during the computing jobs of a) panel, with a reference line at the total installed RAM. c) Ratio of CPU times between the *cw* and *w* cases and (just for optimal path) with and without time-interpolation. d) Ratio of peak RAM allocation of the *cw* to *w* type jobs. For panels a,b,d) both cases are with (filled) and without (empty markers) time-interpolation. The DOF of the time-dependent shortest path problems is displayed on the horizontal axis. e,f) Time series of RAM memory allocation during VISIR execution for *w* and *cw* type jobs, respectively. Black circles (blue lines) refer to runs without (with) time-interpolation of edge weights. Vertical dashed lines separate the main phases of the processing. Both panels refer to the $\nu = 8$ case of a)-d). The processing phase labels are: *ew* (computation of edge-averaged fields); *ed* (edge delays); *gdt* (geodetic track); *opt* (optimal track).

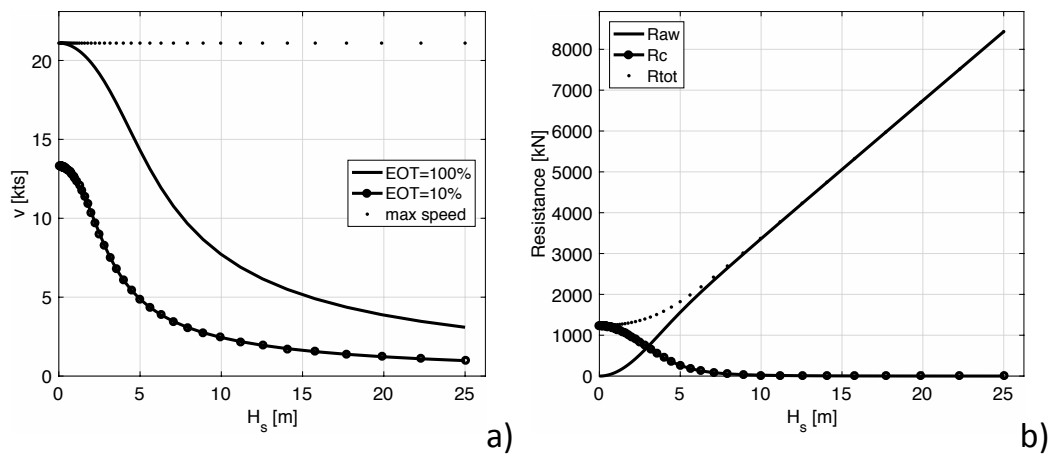

**Figure 4.** Vessel response functions for the parameters given in Tab. 4. a) STW at a constant engine throttle vs. significant wave height $H_s$. Both EOT=100% (solid line) and EOT=10% (line and dots) are displayed. b) Calm water $R_c$, wave-added resistance $R_{aw}$, and their sum $R_{tot}$ as functions of $H_s$.

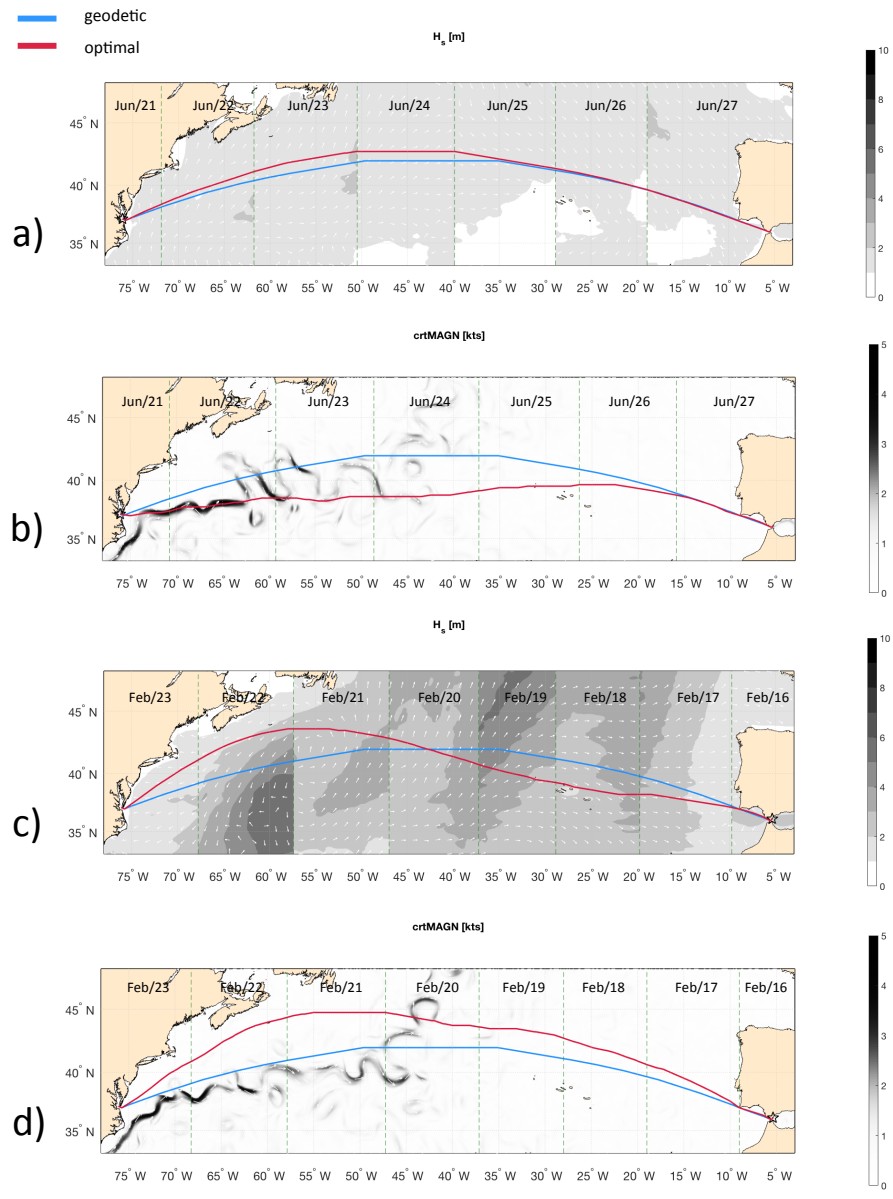

**Figure 5.** Geodetic (blue) and optimal tracks (red) for the USNFK-ESALG route in the presence of different environmental forcings and departure dates: Panels a,c) are of *w* type, $H_s$ is displayed as shading and wave direction as white arrows; Panels b,d) are of *cw*-type and current magnitude is displayed as shading and its direction as white arrows. Departure date is June 21st for a,b) and February 16th for c,d). Departure time is 12Z for all tracks. All panels are split into vertical stripes relative to daily timesteps of the optimal tracks - the interface between stripes is marked by a green dashed line. Summary data reported in Tab. 5.

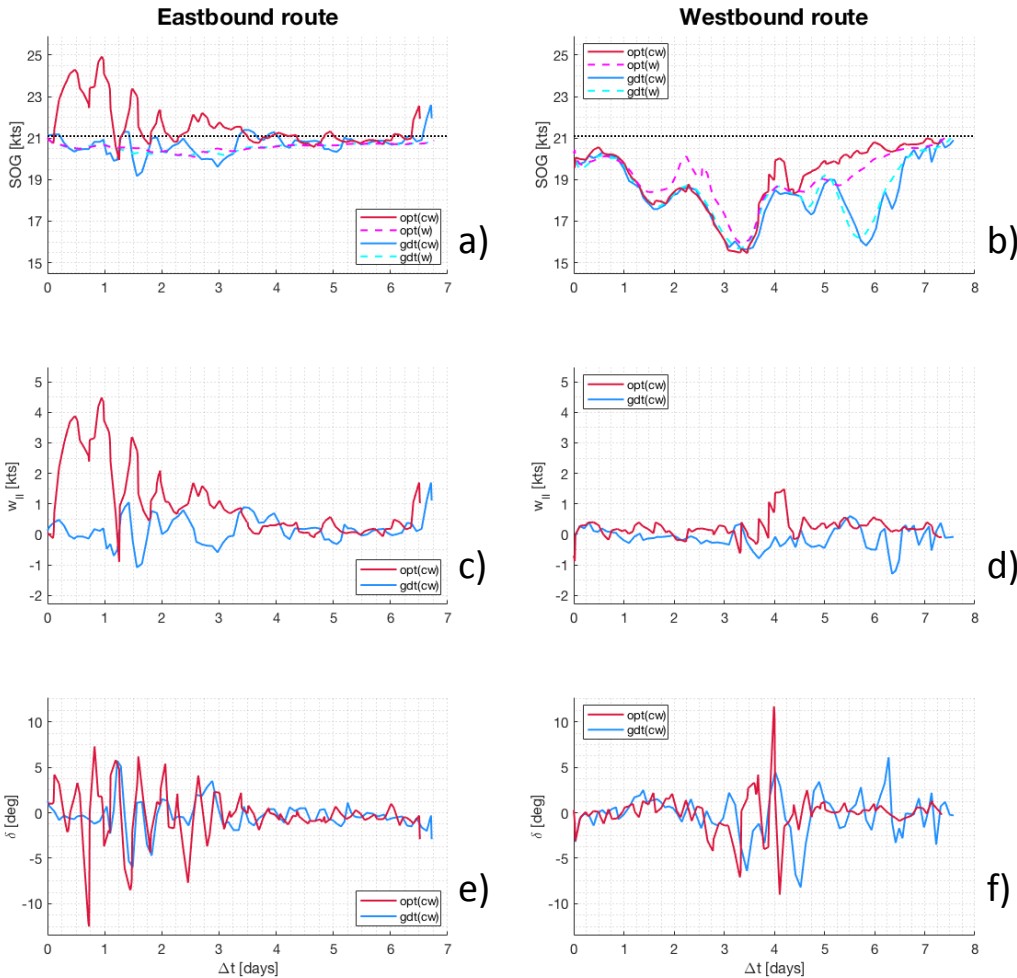

**Figure 6.** Along-route information for both the eastbound (left column) and westbound (right column) crossings of Fig. 5. The first row (Panels a,b) displays the SOG for both optimal and geodetic tracks, for both $w$ and $cw$ cases; the black dotted line is $V_{\max}$ from Tab. 4. The second row (Panels c,d) displays the $w_{\parallel}$ component of the ocean flow, as computed from Eq. 4a. The third row (Panels e,f) displays the angle of attack $\delta$, Eq. 5. The maximum ROT of $\delta$ is $0.8°/\min$ and $0.5°/\min$ for the east- and westbound track, respectively.

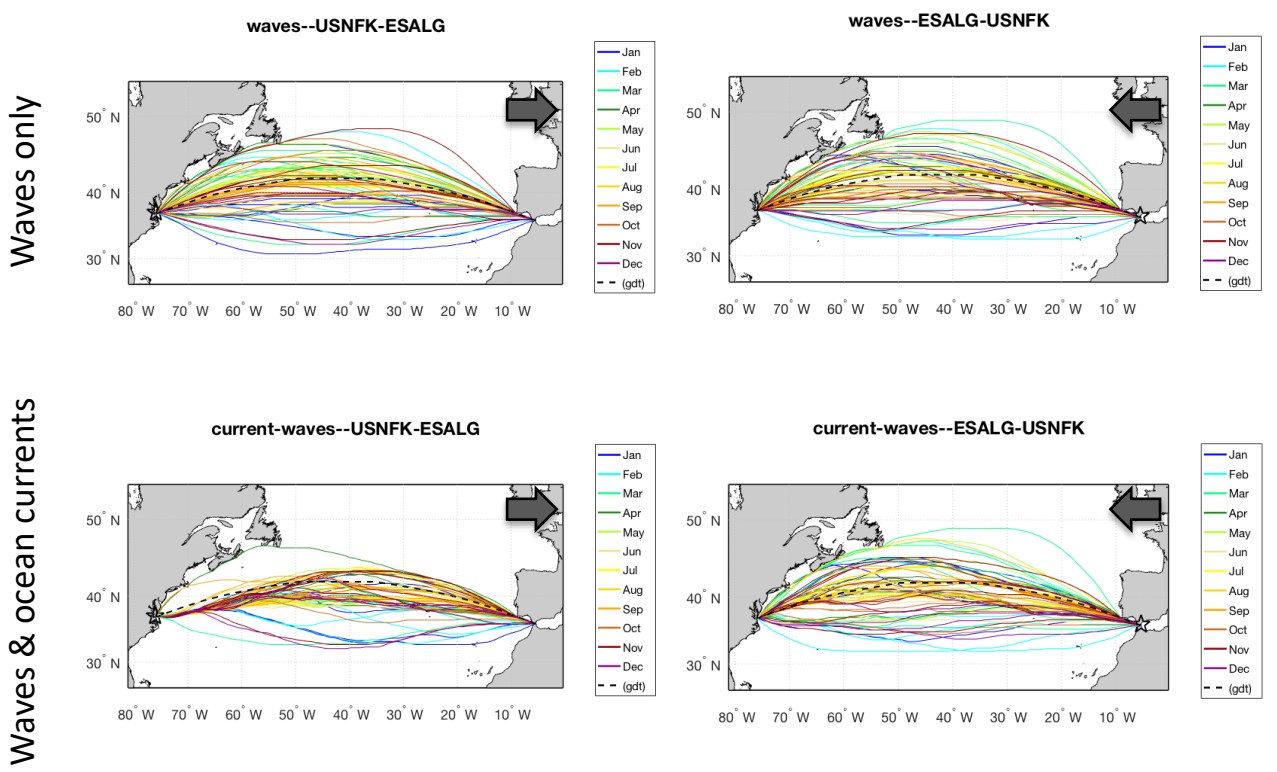

**Figure 7.** Route tracks of the same transatlantic crossing of Fig. 5 during the calendar year 2017. Panels a,b) refer to *w*-type; c,d) to *cw*-type tracks. Both east- (left) and westbound tracks (right) are shown. The geodetic route is displayed as a black dashed line. Animations of the panels are available at https://av.tib.eu/series/560/gmdd+18. For this and other routes, see also the Supplementary Material.

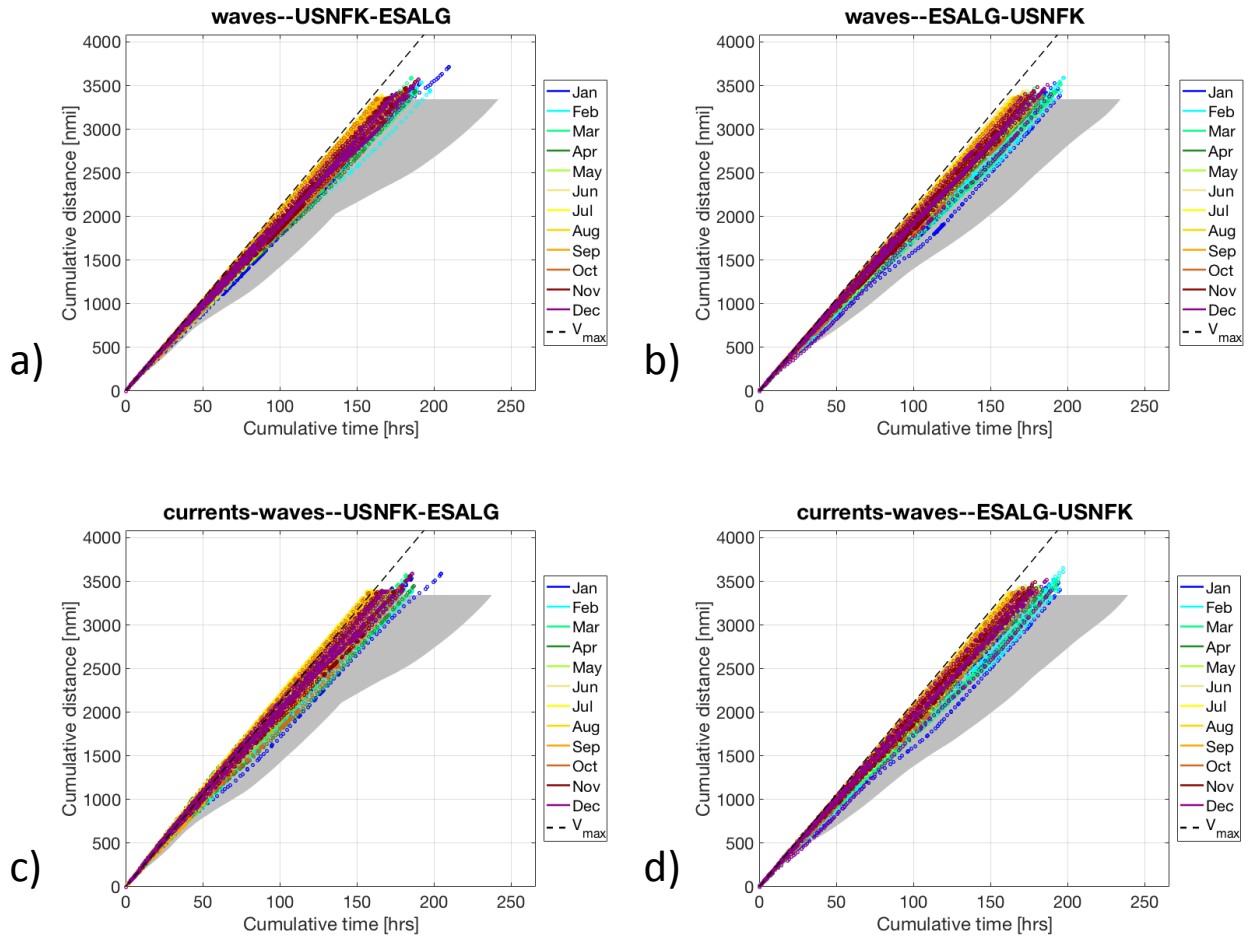

**Figure 8.** Evolution lines for the tracks in Fig. 7: cumulative sailed distance is displayed vs. time elapsed since departure. Each optimal track is displayed with a coloured dot referring to the month of departure as in the legend. The envelope of the geodetic trajectories is shaded in grey. The dashed line refers to sailing at $V_{\mathrm{max}}$.

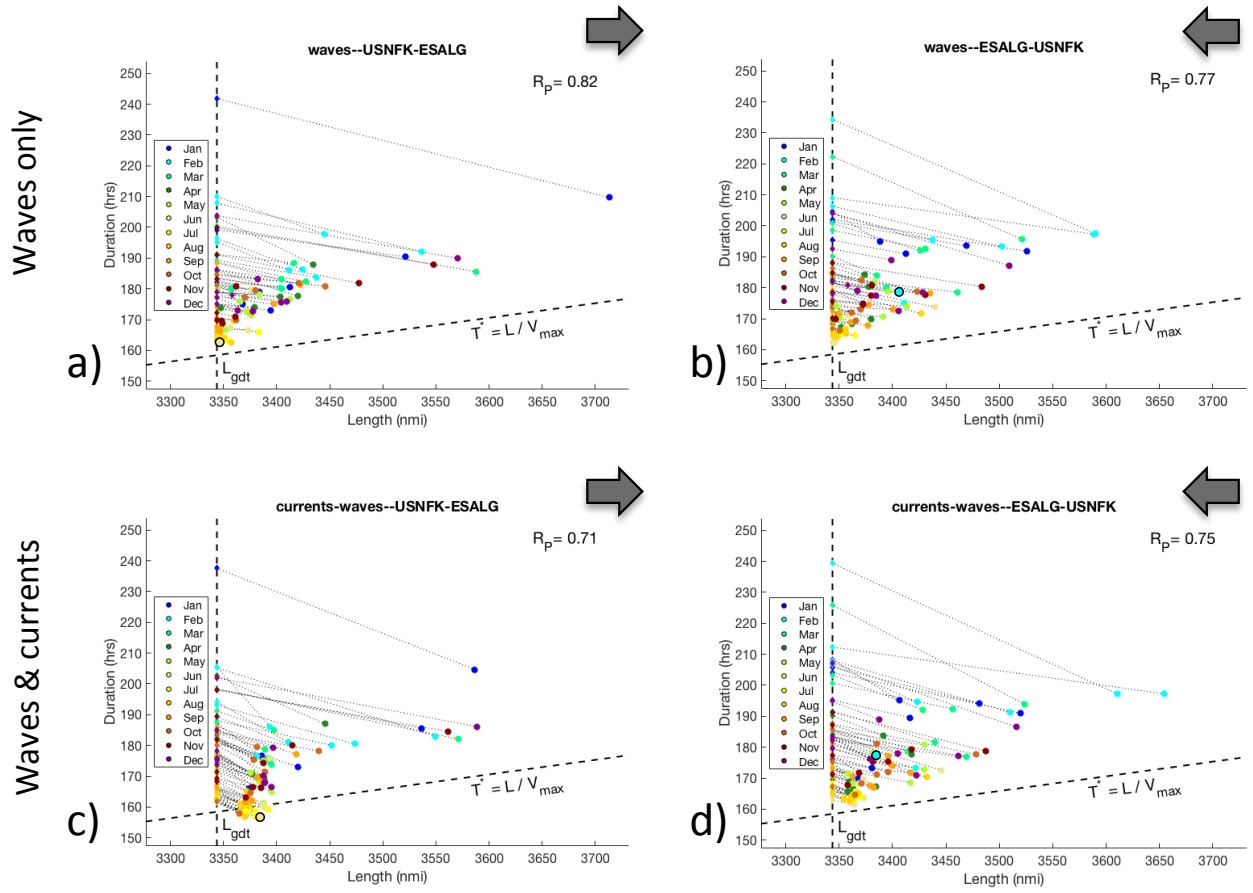

**Figure 9.** Distribution of optimal sailing time $T^*$ vs. length $L$ of the tracks of Fig. 7. For the geodetic tracks, $L = L_{\mathrm{gdt}}$ is a constant. Corresponding optimal track dots $(L, T^*)$ are joined by a light dotted line. The slant dashed line has a slope $V_{\max}^{-1}$. Tracks for *w*-type (Panels a,b) and *cw*-type optimisation (Panels c,d), and for both east- and westbound directions are displayed. Dots of the tracks analysed in Sect. 4.3 are depicted with black edges. The Pearson's correlation coefficient $R_P$ between $T^*$ and $L$ is printed in the top right corner of each panel.

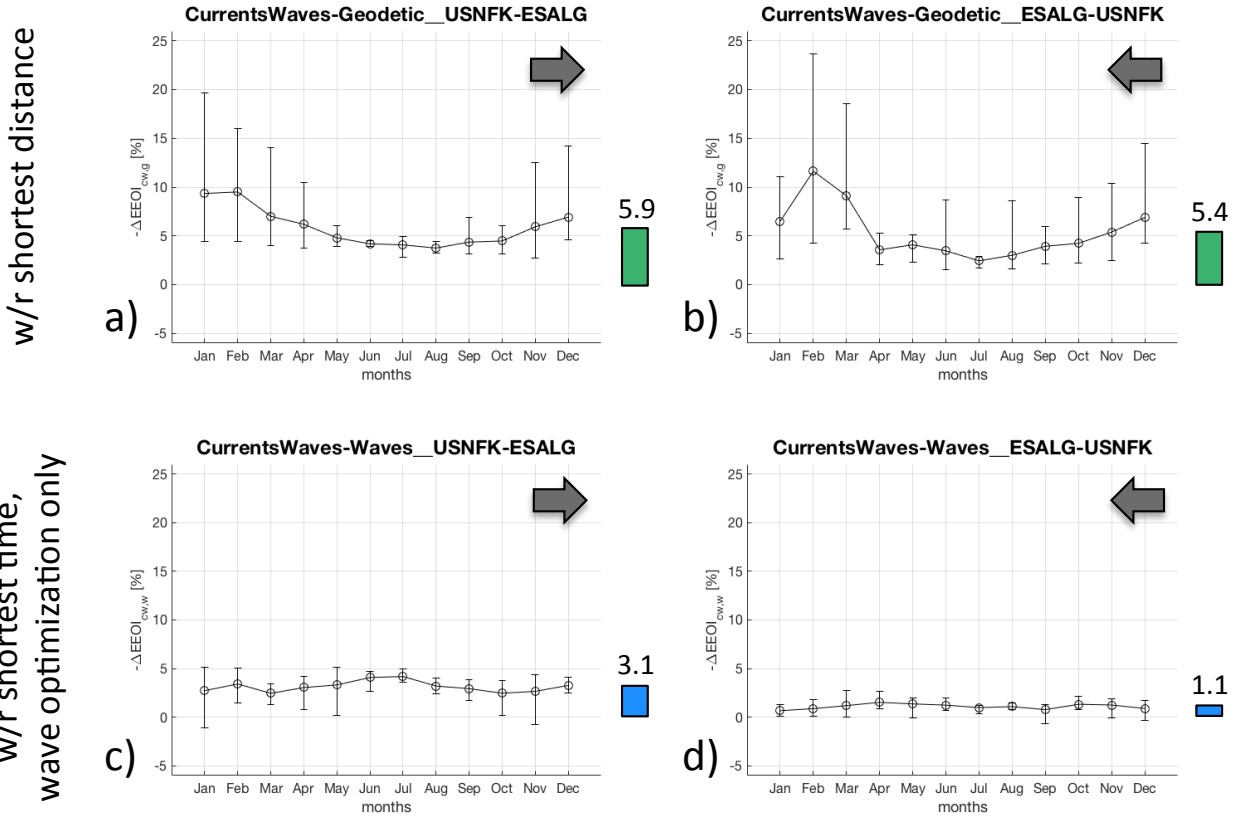

**Figure 10.** EEOI relative savings for the tracks in Fig. 7. The quantity defined in Eq. 18 is computed for optimal tracks of *cw*-type vs. (Panels a,b) the corresponding geodetic tracks and (Panels c,d) vs. corresponding optimal tracks of *w*-type. For each calendar month, the empty circle is positioned at the monthly-average and the error bars span between minimum and maximum value of the (six) routes of that month.

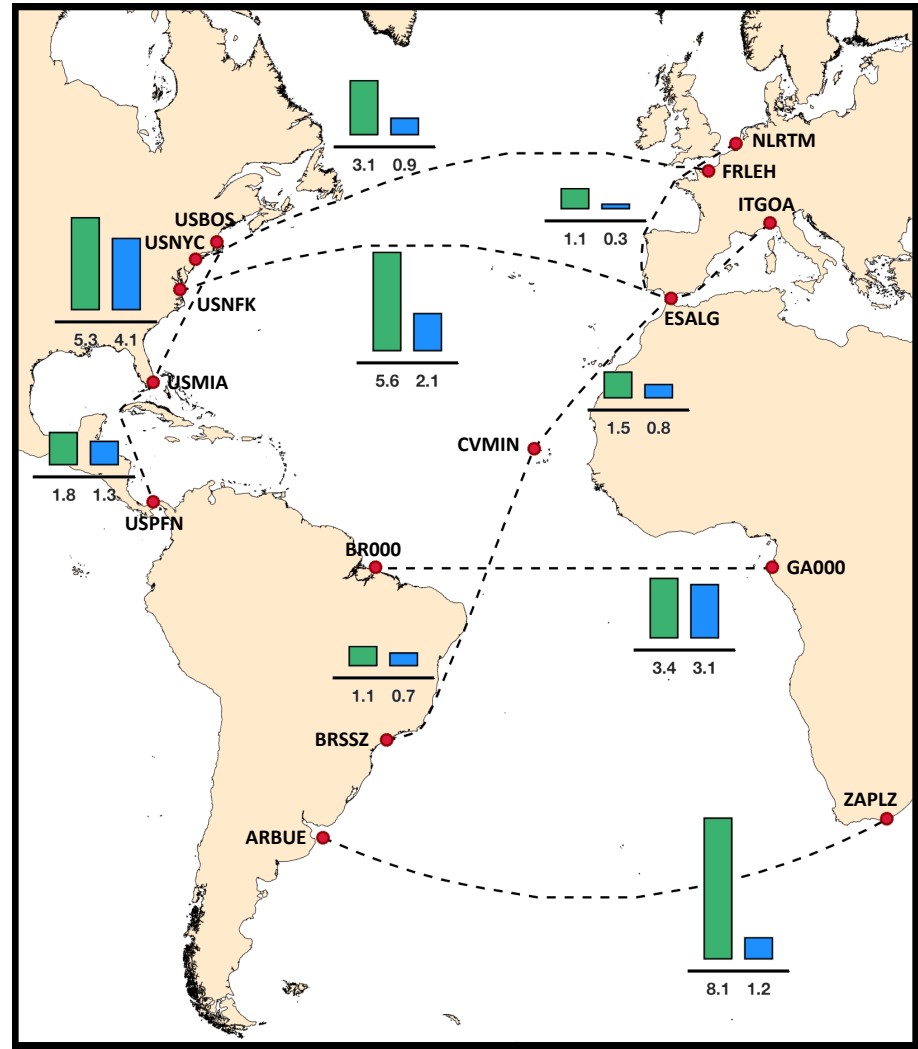

**Figure 11.** Mean relative EEOI savings [%] for several routes in the Atlantic Ocean. The values displayed in the vertical bars refer to the annual average of the mean savings for the return voyages (i.e., mean values along the rows of Fig. 10), sailed along the optimal tracks of *cw* type. The green bars refer to total savings, or $-\Delta(\text{EEOI})_{\text{cw,g}}$, while the blue bars refer to the ocean currents contribution, or $-\Delta(\text{EEOI})_{\text{cw,w}}$.

**Table 1.** Some nautical abbreviations employed in this manuscript.

|  | meaning | units | alternate name |
|---|---|---|---|
| SOG | Speed Over Ground | kn |  |
| STW | Speed Through Water | kn | Pool Velocity (Lo and McCord, 1998) |
| COG | Course Over Ground | ° |  |
| HDG | Heading | ° | Course To Steer |
| ROT | Rate of Turn | °/min |  |
| EOT | Engine Order Telegraph | % | Engine Throttle |

**Table 2.** Summary parameters of benchmark case studies, cf. Fig. 2. Length scale $L_0$ set by the track endpoint distance and time scale $T_0 = L_0/V_{max}$ are employed throughout ($V_{max} = 21.1$ kn). Values in italics correspond to runs without time-interpolation of the edge weights, cf. Sect. 2.4. Values in the last row of each group refer to the analytic solutions.

| units | Input field parameters | | | | Graph parameters | | optimal path metrics | | | |
|---|---|---|---|---|---|---|---|---|---|---|
|  | $\mathcal{R}$ | $g$ | $\Gamma$ | $\Omega$ | $\nu$ | $(\Delta_g)^{-1}$ | $L$ | $\Delta L$ | $T^*$ | $\Delta T^*$ |
|  | $[L_0]$ | $[L_0 \cdot T_0^{-2}]$ | $[T_0^{-1}]$ | $[T_0^{-1}]$ | - | [1/deg] | $[L_0]$ | [%] | $[T_0]$ | [%] |
| wave - (static) | 1/8 | 3.1 | - | - | 2 | 60 | 1.091 | +2.0 | 1.738 | +0.7 |
|  |  |  | - | - | 5 | 25 | 1.079 | +0.9 | 1.725 | +0.3 |
|  |  |  | - | - | 5 | 60 | 1.079 | +0.9 | 1.727 | +0.05 |
|  |  |  | - | - | 10 | 50 | 1.076 | +0.6 | 1.721 | +0.06 |
|  |  |  | - | - | - | - | 1.070 | 0.0 | 1.726 | 0.0 |
| current - (dynamic) | - | - |  |  | 5 | 25 | *1.004* | - | *1.092* | *+6.0* |
|  | - | - |  |  | 5 | 25 | 1.018 | - | 1.056 | +2.5 |
|  | - | - | -0.3 | t - 0.5 | 10 | 50 | 1.014 | - | 1.049 | +1.9 |
|  | - | - |  |  | 10 | 75 | 1.011 | - | 1.047 | +1.6 |
|  | - | - | -0.3 | t - 0.5 | 5 | 100 | 1.010 | - | 1.045 | +1.5 |
|  | - | - |  |  | 5 | 200 | *1.008* | - | *1.086* | *+5.4* |
|  | - | - |  |  | 5 | 200 | 1.007 | - | 1.043 | +1.3 |
|  | - | - |  |  | - | - | - | - | 1.030 | 0.0 |

## Appendix A: List of main changes of VISIR-1.b with respect to VISIR-1.a

The most relevant changes of VISIR-1.b described in this paper are listed in Tab. A1. The list does not include other minor code updates, for which we refer to the release notes of the new model version (cf. "Code and data availability").

**Table 3.** Fit parameters for the data displayed in Fig. 3a. The fit model is $a \cdot x^b + c$. For the optimal path data, $c$ parameter is not fitted.

| | | no T-interp | | with T-interp | |
| --- | --- | --- | --- | --- | --- |
| | units | optimal path | total job | optimal path | total job |
| $a$ | s | $9.9 \cdot 10^{-8}$ | $4.7 \cdot 10^{-10}$ | $2.6 \cdot 10^{-6}$ | $1.2 \cdot 10^{-7}$ |
| $b$ | – | 1.07 | 1.42 | 1.01 | 1.18 |
| $c$ | s | - | 52 | - | 60 |
| rmse | s | 3.9 | 15.6 | 3.3 | 24.8 |

**Table 4.** Database of vessel propulsion parameters and principal particulars used in this work. The values of $\Delta$ (ballast – scantling) and the maximum cargo capacity (2 500 TEUs) are not used in the computations and are provided just for the sake of reference.

| Symbol | Name | Units | Value(s) |
| --- | --- | --- | --- |
| SMCR | optimal maximum continuous rating power | kW | 19 164 |
| $V_{\mathrm{max}}$ | top design speed | kn | 21.1 |
| $L_{\mathrm{wl}}$ | length at waterline | m | 210 |
| $B_{\mathrm{WL}}$ | beam (width at waterline) | m | 30 |
| $T$ | draught | m | 11.5 |
| $T_{\mathrm{R}}$ | ship natural roll period | s | 21.2 |
| $C_T$ | drag coefficient | - | $\gamma_q \mathrm{STW}^q$ |
| $q$ | exponent in $C_T$ | - | 2 |
| $\Delta$ | displacement | m$^3$ | 21 600 – 45 600 |
| DWT | deadweight | t | 33 434 |

## Appendix B: Note on manoeuvring and actuation

In order to head as prescribed by the optimal track, the ship has to be manoeuvred (e.g. acting on rudder and/or lateral thrusters, Bertram (2000)). The rudder is handled via a hydraulic device that converts pressure into a mechanical action to move the rudder[21]. In order to implement the prescribed EOT, the high level order from the control bridge is transmitted through poten-
5 tiometers[22] to the main engines (and possibly also to other components of the propulsion system such as clutches, gearbox, controllable pitch propeller, cf. Harvald (1992)).

Motions of the bottom layer (rudder, main engine), as related to electro-mechanical devices, should occur on a much shorter timescale (probably seconds to a few minutes) than the top level controls needed for implementing the optimal track (requiring changes in the order of minutes, cf. $ROT_M$ in Tab. 7, to hours, cf. Fig. 6). Thus, a routing system must ensure that the top
10 level control requires feasible manoeuvers (e.g. in Sect. 4.3.2 we check that maximum vessel Rate of Turn $ROT_M$ is in an

---

[21]https://www.wartsila.com/encyclopedia/term/rudder-actuator
[22]https://www.kwantcontrols.com/product/systems/integrated-telegraph-system/

**Table 5.** Route length $L$ (or $L^*$ for geodetic tracks) and optimal duration $T^*$ (or $T$ for geodetic tracks) for tracks in Fig. 5-6. $\Delta T_g$ is the relative duration change with respect to the geodetics; $\Delta T_w$ with respect to $w$-type optimal tracks. On the WGS-84 geoid, the length of the arc of GC between the endpoints is 3332.60 nmi, i.e. the numerical solution overestimates it by 0.3%. In a still ocean (no currents nor waves) the numerical geodetic would be sailed in 158 :28 :28 hrs by a vessel with $V_{max}$ as in Tab. 4. The second header line of the $-\Delta$EEOI columns specifies the type of tracks as in Eq. 18.

| track direction | track type | forcings | $L$ (or $L^*$) | $T^*$ (or $T$) | $\Delta T_g$ | $\Delta T_w$ | $-\Delta$EEOI | |
|---|---|---|---|---|---|---|---|---|
| | | | nmi | hh : mm : ss | % | % | $\beta, g$ | cw,w |
| Eastbound (2017-06-21) | geodetic | w | 3343.81 | 162 :48 :34 | - | - | - | - |
| | | cw | | 161 :43 :10 | - | - | - | |
| | optimal | w | 3346.46 | 162 :44 :13 | 0.04 | - | 0.12 | 4.75 |
| | | cw | 3384.02 | 156 :44 :48 | 3.07 | 3.68 | 4.23 | |
| Westbound (2017-02-16) | geodetic | w | 3343.81 | 181 :25 :18 | - | - | - | - |
| | | cw | | 182 :44 :57 | - | - | - | |
| | optimal | w | 3405.85 | 178 :26 :41 | 1.64 | - | 3.43 | 0.12 |
| | | cw | 3384.69 | 177 :06 :52 | 3.08 | 0.75 | 4.25 | |

acceptable range; other feasibility criteria are defined in IMO (2002)). If this condition is satisfied, it should be possible, for the sake of computation of the optimal track, to safely ignore the temporal dynamics of the underlying actuation level (Techy, 2011). On the other hand, if the actuator time scale were comparable to the time over which heading and EOT changes should take place, the hypothesis of top-bottom level separation would be invalid. We presume that this is much less likely to occur in open-sea navigation (which is the subject of the present manuscript) than, for example, during harbour operations. However, on board data would be needed for a thorough assessment of this issue.

## Appendix C: Note on alternative graph meshes

Following Mannarini et al. (2016a), we took into consideration the fact that the VISIR graph grid may need to be redesigned, e.g. by reducing the density of gridpoints in open seas through the use of a nonuniform mesh. An adaptive refinement mesh (Berger and Colella, 1989) or unstructured mesh limiting the minimum angle (Shewchuk, 2002) could be another option. This would reduce the number of open-ocean edges, thereby reducing RAM allocation (cf. Sect. 3.2.2) and speeding up the computation of the shortest path.

In any case, to ensure navigation safety , the intersection between graph arcs and shoreline (Sect. 2.3) needs to be verified, irrespectively of the grid resolution or structure. In fact, even if the mesh is built via a tessellation, intersection with islands and boundary elements smaller than mesh elements should be checked (Legrand et al., 2000). For a graph of higher order of connectivity ($\nu \gg 1$) this is even more challenging. Such a check on shoreline intersection can easily represent a significant computational cost (De Berg et al., 1997). In order to perform it effectively, it is crucial to be able to find indexes of graph

**Table 6.** Database of harbours. Coordinates refer to the graph grid point selected by VISIR. Wherever available, GRT is the annual through-put for the year 2016 from Lloyd's (2018) and is used for sorting the entries. The other harbours are sorted alphabetically following the International Seaport Code.

| Code | Name | Lat [°N] | Lon [ °E] | GRT [TEU] |
|---|---|---|---|---|
| NLRTM | Rotterdam | 52.000 | 4.000 | 12,385,168 |
| USNYC | New York | 40.500 | -73.875 | 6,251,953 |
| ESALG | Algeciras | 36.125 | -5.375 | 4,761,428 |
| BRSSZ | Santos | -24.125 | -46.375 | 3,393,593 |
| USPFN | Panama (Colón) | 9.375 | -80.000 | 3,258,381 |
| USNFK | Norfolk (Virginia) | 37.125 | -76.125 | 2,655,705 |
| ITGOA | Genoa | 44.375 | 8.875 | 2,297,917 |
| ARBUE | Buenos Aires | -36.250 | -55.500 | - |
| BR000 | Brazil's end of Equator | 0.000 | -48.000 | - |
| CVMIN | Mindelo | 16.875 | -25.125 | - |
| FRLEH | Le Havre | 49.500 | 0.000 | - |
| GA000 | Gabon's end of Equator | 0.000 | 9.250 | - |
| USBOS | Boston | 42.375 | -70.875 | - |
| USMIA | Miami | 25.750 | -80.000 | - |
| ZAPLZ | Port Elizabeth | -34.000 | 25.750 | - |

elements next the shoreline. On a regular grid this operation can be carried out in $\mathcal{O}(M)$ time ($M$ is the number of shoreline elements), irrespectively of the size of the maritime domain (and we exploited this in the *i)* step of the algorithm described in Sect. 2.3). Instead, on a random or not regular mesh, a $\mathcal{O}(M \cdot n)$ time would be required by a linear search ($n$ is here either the number of nodes or arcs of the graph). To speed up the search on a not regular mesh, a preliminary node indexing can be computed. With a *k-d* tree, an additional $\mathcal{O}(n\log(n))$ time for tree construction and, on average, $\mathcal{O}(M \cdot \log(n))$ for querying would be needed (Bentley, 1975). This is in excess of the $\mathcal{O}(M)$ estimate for corresponding step (cf. *i)* in Sect. 2.3) in the present VISIR graph creation algorithm.

Thus, at this stage we still use a regular grid which enables a relatively quick and easy graph computation at the cost of a longer path computing time. This is not critical, given the non-operational functioning of VISIR for the present exercise. In future model versions, also depending on coding options, domain, and type of application, we may reconsider this choice.

## Appendix D: Note on model performance comparison

Since the VISIR solution is based on Dijkstra's algorithm, it is not just guaranteed to be exact, however its performance (for a given route and vessel departure date) is stable over subsequent runs. This is a difference with evolutionary (EA) and, generally speaking, with heuristics-based algorithms. For that class of algorithms, both the quality and the computational cost of the

**Table 7.** Database of routes. $L_\mathrm{g}$ is the length of the geodetic track on the graph. $\Delta$ is a shortcut for the EEOI saving. The $\langle \cdot \rangle$ operator denotes the annual mean, the $\triangleleft \cdot \triangleright$ the mean annual value of the standard deviation. Corresponding values are given in %. The second header line specifies the type of tracks. The other columns contain: the number of tracks $N_E$ with intentional speed reduction and the maximum % fraction of track waypoints affected ($W_P$) – for the $w$-type this figure is always 0 but for the ZAPLZ-ARBUE route, where it reads 1(0.4); the maximum rate of turn $ROT_M$ (°/min); the number of non-FIFO edges $\overline{F}$ (neither of them is along the optimal track); the Pearson coefficient $R_P$ between $T^*$ and $L$. The DOF varies from more than $5.4 \cdot 10^8$ of the ARBUE-ZAPLZ to about $2.5 \cdot 10^7$ of the USBOS-USMIA.

| port #1 | port #2 | $L_\mathrm{g}$ [nmi] | $\langle -\Delta\rangle$ | $\triangleleft -\Delta\triangleright$ | $\langle -\Delta\rangle$ | $\triangleleft -\Delta\triangleright$ | $N_E(W_P)$ | $ROT_M$ | $\overline{F}$ | | $R_P$ | |
|---|---|---|---|---|---|---|---|---|---|---|---|---|
| | | | cw,g | | cw,w | | cw | cw | w | cw | w | cw |
| ARBUE | ZAPLZ | 3872.13 | 8.0 | 5.4 | 1.4 | 1.3 | 0 | 2.3 | 0 | 47 | 0.73 | 0.67 |
| ZAPLZ | ARBUE | | 8.2 | 4.0 | 1.1 | 0.7 | 1 (0.6) | 3.4 | | | 0.50 | 0.51 |
| BR000 | GA000 | 3442.18 | 1.8 | 0.7 | 4.3 | 1.2 | 0 | 0.7 | 0 | 0 | 0.55 | -0.05 |
| GA000 | BR000 | | 5.0 | 1.8 | 1.8 | 1.2 | 0 | 0.8 | | | 0.38 | 0.05 |
| USNFK | ESALG | 3343.81 | 5.9 | 3.3 | 3.2 | 1.3 | 0 | 3.0 | 0 | 24 | 0.83 | 0.71 |
| ESALG | USNFK | | 5.4 | 4.1 | 1.2 | 1.0 | 0 | 2.3 | | | 0.77 | 0.75 |
| USNYC | FRLEH | 3076.73 | 2.7 | 2.3 | 0.8 | 2.3 | 2 (3.9) | 14.8 | 0 | 26 | 0.62 | 0.65 |
| FRLEH | USNYC | | 3.4 | 2.5 | 0.6 | 2.3 | 1 (2.0) | 2.0 | | | 0.66 | 0.67 |
| BRSSZ | CVMIN | 2852.16 | 1.2 | 0.4 | 0.2 | 0.7 | 0 | 16.2 | 0 | 0 | 0.74 | 0.36 |
| CVMIN | BRSSZ | | 1.04 | 0.4 | 1.3 | 0.7 | 0 | 1.9 | | | 0.62 | 0.32 |
| CVMIN | ITGOA | 2406.48 | 1.4 | 0.5 | 0.4 | 0.4 | 0 | 14.8 | 0 | 0 | 0.66 | 0.39 |
| ITGOA | CVMIN | | 1.6 | 0.7 | 1.1 | 0.4 | 0 | 2.4 | | | 0.54 | 0.51 |
| NLRTM | ESALG | 1334.51 | 1.2 | 1.4 | 0.6 | 1.2 | 0 | 3.7 | 0 | 0 | 0.95 | 0.93 |
| ESALG | NLRTM | | 1.1 | 1.2 | 0.2 | 1.6 | 0 | 16.8 | | | 0.92 | 0.88 |
| USMIA | USPFN | 1171.74 | 2.0 | 0.8 | | 0.9 | 0 | 2.3 | 0 | 2 | 0.75 | 0.19 |
| USPFN | USMIA | | 1.7 | 0.5 | 2.7 | 0.7 | 0 | 14.6 | | | 0.71 | 0.18 |
| USBOS | USMIA | 1146.91 | 5.4 | 1.7 | 1.0 | 0.9 | 0 | 1.3 | 2 | 6 | 0.82 | 0.47 |
| USMIA | USBOS | | 4.9 | 1.4 | 6.9 | 1.9 | 0 | 19.6 | | | 0.85 | -0.06 |

**Table A1.** List of main code changes of VISIR-1.b with respect to VISIR-1.a version, with indication of their use within this manuscript.

| object | type of change | Ref. within this paper |
|---|---|---|
| use of ocean currents | new feature | Sect. 2.2 |
| graph generation | generalisation | Sect. 2.3 |
| graph resolution | generalisation | Sect. 2.3 |
| time interpolation of edge weights | new feature | Sect. 2.4 |
| parametric roll threshold condition | generalisation | Sect. 2.5.2 |
| input model fields | generalisation | Sect. 4.1 |

solution may vary over subsequent runs, as they are driven by random effects. The issue of randomness can be mitigated by statistical averaging over many simulations. However, a more fundamental issue is that, as clearly stated in Eiben et al. (2003), the performance of an EA should be assessed in terms of both efficiency (CPU time) and effectiveness (quality of the solution). Furthermore, even for a specific EA and EA implementation, performance may vary with tuning. Tuning refers to specifying values for the algorithm parameters, such as the "mutation rate". Tuning may affect both EA performance and robustness (Eiben et al., 2003). Apart from the particular features of EA, comparing the performance of VISIR with other ship routing systems is also hampered by the facts mentioned in Sect. 1.1. These need to be overcome in dedicated collaborative efforts., as we did in (Mannarini et al., 2018, in review). We are open to replicating that approach for EA-based ship routing models, e.g. the ant-colony algorithm described in Tsou and Cheng (2013) or the multi-objective EA reported in (Szlapczynska, 2015).