# Peer review of "VISIR-1.b: ocean surface gravity waves and currents for energy efficient navigation"

_Geoscientific Model Development, 2018_

## Referee Comment (RC1) · Anonymous Referee #1 · 19 Mar 2019

This is a reasonably good paper that describes a new version of a ship-routing model. The original model was published in GMD, so the subject matter has already been judged to fall within the scope of the journal. The manuscript assesses the impact of waves and currents on transatlantic crossings, and calculates energy efficiency savings that seem impressive.

Major comments:

1) The term "waves" is used throughout the manuscript, but it is never properly defined. The ocean supports a wide variety of wave motions, both internal and at the surface, including gravity waves, Rossby waves, Kelvin waves, Poincare waves, acoustic waves, etc. I believe the manuscript is referring exclusively to surface gravity waves, but this needs to be stated. In the equivalent atmospheric problem of aircraft routing, "waves"

usually refers to Rossby waves in the jet stream, and the wave structure (in other words, the u(x,y,t) and v(x,y,t) velocity field associated with the wave) is used in the calculation of the fastest route. I presume that the flow perturbation associated with the surface gravity waves in the current manuscript is not being used like this, but rather that the waves are being treated as areas of turbulence to be avoided. However, this wasn't entirely clear to me and deserves to be clarified.

2) The manuscript is missing a discussion on whether the ship-routing model is intended for operational use or just for research purposes. More generally, it is missing a discussion on how ships are currently routed operationally: are the tracks optimal in some sense? If so, who calculates the optimal routes, and using what model? This is particularly relevant to interpret the energy efficiency gains calculated in the manuscript.

3) I generally found the manuscript difficult to read and understand, mostly because of the poor quality of English usage throughout. This problem could and should be fixed by calling on a native English speaker or professional proof-reading service.

Minor comments:

Page 1, line 20: "which capacity" -> "whose capacity".

Page 2, line 2: please define "dead reckoning".

Page 4, lines 15-16: what are the manoeuvrability and actuation issues that arise, and what are the consequences of not considering them?

Page 7, line 17: "preliminary" -> "preliminarily".

Page 7, line 29: the final sentence makes no sense.

Page 10, line 14: "anthropic" -> "anthropogenic".

Page 26, line 25: please specify which version of Matlab.

[Figure]

Page 37, Figure 5: the geodetic curves look piecewise linear (i.e. local geodetics between waypoints) rather than continuous - why?

Page 39, Figure 7: the captions refer us to an external website for the animations. I think they should probably refer us to the supplementary material instead.

Page 39, Figure 7: "oncean" -> "ocean" in the ordinate label.

---

## Referee Comment (RC2) · Anonymous Referee #2 · 1 Apr 2019

General assessment The paper is a through one. There are very few papers on ship weather routing covering so many aspects of this optimization problem and doing it with so much detail. I particularly appreciate:

- the time interpolation - I agree that it may bring significant benefits for drastically changes in the subsequent weather forecasts,

- using bathymetric database with detailed real data,

- detailed results and analysis of time savings attributed to exploitation of waves and currents.

My specific comments are few – I provide them below.

[Figure]

Specific comments

1. 'We use throughout this manuscript the words "track" or "trajectory" for indicating a set of waypoints joining two given endpoints or harbours, in relation to departure on a given date, and the words "route" or "crossing" when there is no reference to a specific departure date.'

While 'track' is perfectly acceptable here, I suggest replacing 'trajectory' with some other word (e.g. path). The word 'trajectory' is usually used in control and robotics with a different meaning: it involves greater accuracy (manoeuvrability and actuation issues), especially for obstacle avoidance or collision avoidance purposes. A "trajectory" between two harbours does not make sense.

2. Regarding section 2.3: an alternative approach would be to use varying resolution of a graph – the nodes can be placed with larger resolution in coastal areas and with lower resolution at open waters. I suggest commenting on the those two possible approaches to this problem and explaining why you choose the one with additional intersection check.

3. Regarding section 2.5.2: 'Edges which, for a given EOT, violate stability are pruned before the shortest path algorithm is run. This way, it is ensured that the optimal track preserves vessel intact stability.' Based on the above description, I am not sure if this approach is correct. In presence of coastline, shallows etc. the exact time at which an edge will be transited cannot be know exactly prior to running the algorithm. Even for open ocean, avoiding a cyclone may cause a delay resulting in reaching a certain graph node much later, thus making all prior assumptions inaccurate. Therefore, in my opinion the edges' weights should be verified dynamically during the algorithm run instead of pruning the edges before the run.

4. While I appreciate the computational complexity analysis based on RAM allocation data, I would also hope for assessing computational time and space based on the algorithm itself. I agree that it is a hard task for complex algorithms, but still some

analysis could be made, at least for the worst case. It would also be interesting to compare the computational time with that of a non-deterministic approach (there are multiple meta-heuristics available, including Evolutionary Multi-objective Optimization, Ant Colony Optimization etc.).

5. I agree with the authors that the paper would further benefit from a more realistic modeling of speed loss in waves and wind. I encourage them to include such modelling in their research.

---

## Author Comment (AC1) · 15 Apr 2019

A web application for interactive exploration of the tracks of manuscript's Sect.4 and Supplementary Material is available at http://www.atlantos-visir.com/. The application allows zooming-in optimal tracks, checking their capacity in landmass avoidance and getting the EEOI savings compared to the least-distance track. Finally, the tracks can be filtered by calendar month, easing to visualize the seasonal dependence.

Acknowledgements: Nadia Pinardi (University of Bologna), Matteo Scuro (CMCC)

---

## Author Comment (AC2) · 20 May 2019

**General assessment**

*This is a reasonably good paper that describes a new version of a ship-routing model. The original model was published in GMD, so the subject matter has already been judged to fall within the scope of the journal. The manuscript assesses the impact of waves and currents on transatlantic crossings, and calculates energy efficiency savings that seem impressive.*

−AUTHORS' RESPONSE:

We thank the Referee for his/her time and comments on our manuscript: They definitively contributed to improve it. In this document, we report Referee's text in italics and our replies as a normal text, distinguishing wherever needed our response from the manuscripts parts involved by changes. All references to sections, equations, figures, and tables are relative to the submitted gmd-2018-292 manuscript.

**Specific comments**

**1** - *The term "waves" is used throughout the manuscript, but it is never properly defined. The ocean supports a wide variety of wave motions, both internal and at the surface, including gravity waves, Rossby waves, Kelvin waves, Poincare waves, acoustic waves, etc. I believe the manuscript is referring exclusively to surface gravity waves, but this needs to be stated. In the equivalent atmospheric problem of aircraft routing, "waves" usually refers to Rossby waves in the jet stream, and the wave structure (in other words, the $u(x, y, t)$ and $v(x, y, t)$ velocity field associated with the wave) is used in the calculation of the fastest route. I presume that the flow perturbation associated with the surface gravity waves in the current manuscript is not being used like this, but rather that the waves are being treated as areas of turbulence to be avoided. However, this wasn't entirely clear to me and deserves to be clarified.*

–AUTHORS' RESPONSE:

In this manuscript we refer to and employ numerical fields of ocean surface gravity waves.

This is in fact the kind of wave motion contributing to so called "wave added resistance" of a vessel. Such resistance is in physical terms a force leading to involuntary vessel
* * *
speed loss in waves. This force is traditionally distinguished into a radiation (energy dissipated due to vessel heave and pitch motions) and a diffraction component (energy dissipated by the hull to deflect incoming waves, short with respect to vessel length). As in Mannarini et al. (2016a), diffraction is neglected in the present parametrization (Sect.2.5.1). However, for both radiation and diffraction motions, the relevant wave length scale is set by vessel length (which is up to a few hundred m).

Furthermore, in VISIR we do account also for vessel intact stability (Sect.2.5.2), which sets a time scale given by the vessel natural roll period (usually up to about 20 s, or more than 0.05 Hz). The CMEMS wave fields employed in VISIR stem from the Météo-France model, which considers a wave spectrum discretized into 24 directions and 30 frequencies in the $[0.035 - 0.58]$ Hz range[1]. Classically, this is the realm of ocean surface gravity waves (Munk, 1951).

–MANUSCRIPT PARTS INVOLVED:
We propose to make changes in following parts:

- Title, to be changed into "VISIR-1.b: ocean surface gravity waves and currents for energy efficient navigation".

- Sect.1.1. Add specification of what ocean waves are considered.

- Sect.4.1.4. Add information that the wave spectrum is discretized into 24 directions and 30 frequencies starting from 0.035 Hz to 0.58 Hz[2], comparing to typical vessel natural roll frequencies.

**2** - *The manuscript is missing a discussion on whether the ship-routing model is*
* * *
[1]https://bit.ly/2KWCHYL
[2]https://bit.ly/2KWCHYL

*intended for operational use or just for research purposes. More generally, it is missing a discussion on how ships are currently routed operationally: are the tracks optimal in some sense? If so, who calculates the optimal routes, and using what model? This is particularly relevant to interpret the energy efficiency gains calculated in the manuscript.*

–AUTHORS' RESPONSE:

VISIR can be used either with analysis or forecast environmental fields, since this is not constrained by any of the equations of Sect.2. Thus, VISIR can help for both assessment of past tracks (as in the present work) or prediction of optimal ones (as actually done in the operational system for the Mediterranean Sea described in Mannarini et al. (2016b)).

In the mapping exercise in the Atlantic Ocean included in the present manuscript, for the reason discussed in Sect.4.5. (duration of the transatlantic crossing exceeding maximum lead time of wave forecasts) and the fact that an operational system was not required by the funding project, we resorted to analysis fields. We think this approach can be useful for ex-post assessments of energy efficiency savings. To this end, the main limitation of our approach is the parametrization of speed loss in surface gravity waves (cf. Question 1) above and Sect.2.5.1), which suffers from still large uncertainties in the literature for the wave added resistance (Bertram and Couser, 2014). Energy efficiency gains resulting from VISIR refer to comparison of the least-time to the orthodromic path and their entity also depends on the amount of speed loss in waves.

Concerning the degree of optimization of actually sailed ship tracks, this is an open research question. Weather ship routing systems are used both offshore and onboard for planning, but the final decision is up to the shipmaster (Fujii et al., 2017). Furthermore, route planning may involve sensitive commercial information that a ship operator will not easily share. Thus, the extent to which a ship track is optimized is not always

publicly known.

Nevertheless, we recently addressed this question by comparing VISIR optimal tracks vs. reported ship tracks per AIS (Automated Identification System) data, for a route in the Southern Ocean (Mannarini et al., 2019). By computing both spatial and temporal discrepancies between VISIR and AIS tracks, we could infer that optimization likely took place in several but not all tracks. While the method by Mannarini et al. (2019) is still in its infancy, we believe its extension to a larger statistics could contribute to shed light on questions like the one posed in this Referee's comment.

–MANUSCRIPT PARTS INVOLVED:
The above discussion can be added to Sect.4.5.

**3** - *I generally found the manuscript difficult to read and understand, mostly because of the poor quality of English usage throughout. This problem could and should be fixed by calling on a native English speaker or professional proof-reading service.*

–AUTHORS' RESPONSE:
Thanks for feedback. We have already contacted a professional proof-reading service for reviewing the final version of the manuscript.

–MANUSCRIPT PARTS INVOLVED:
Whole manuscript.

**Minor Comments**

*1) Page 1, line 20: "which capacity" -> "whose capacity".*

−AUTHORS' RESPONSE:
Thanks, this will be fixed.

*2) Page 2, line 2: please define "dead reckoning".*

−AUTHORS' RESPONSE:
Dead reckoning refers to the computation of a vessel's position by means of its position at a past time and advancing it, based upon estimated speed and course over elapsed time. In the work by Richardson (1997), ship drift was defined as the difference of the velocity vector between two position fixes (by means of some reliable method) and the velocity vector resulting from dead reckoning. In Meehl (1982) a similar definition of ship drift was given, with the specification that dead reckoning is to be computed 24 h in advance of the position fix.

−MANUSCRIPT PARTS INVOLVED:
Above answer will be added in the Introduction.

*3) Page 4, lines 15-16: what are the manoeuvrability and actuation issues that arise, and what are the consequences of not considering them?*

−AUTHORS' RESPONSE:
VISIR computes heading and fraction (EOT) of maximum engine power to be held along an optimal ship track.

In order to head as prescribed by the optimal track, the ship has to be manoeuvred (e.g. acting on rudder and/or lateral thrusters). Action on rudder is realized through some hydraulic device converting pressure into a mechanical action to move the rudder[3].
In order to implement prescribed EOT, the high level order from the control bridge is
* * *
[3]https://www.wartsila.com/encyclopedia/term/rudder-actuator

transmitted through potentiometers[4] to the main engines (and possibly also to other components of the propulsion system such as clutches, gearbox, controllable pitch propeller, cf. Harvald (1992)).

Motions of the bottom layer (rudder, main engine), being related to electro-mechanical devices, should occur on a much shorter timescale (probably seconds to a few minutes) than the top level controls needed for following the optimal track (requiring changes in the order of minutes, cf. $ROT_M$ in Tab.7, to hours, cf. Fig.6). Thus, a routing system must ensure that the top level control requires feasible manoeuvers (e.g. in Sect.4.3.2 we check that maximum vessel Rate of Turn $ROT_M$ is in an acceptable range; other feasibility criteria are defined in IMO (2002)). If this condition is given, it should be possible, for the sake of computation of the optimal track, to safely ignore the temporal dynamics of the underlying actuation level (Techy, 2011). If instead actuator time scale were comparable to the time over which heading and EOT changes should take place, the hypothesis of top-bottom level separation would break down. We suppose that this is much less likely to occur in open-sea navigation (object of the present manuscript) than e.g. during harbour operations. However, on board data would be needed for a thorough assessment of this issue.

–MANUSCRIPT PARTS INVOLVED:
Sect.2.1.

*4) Page 7, line 17: "preliminary" -> "preliminarily".*

–AUTHORS' RESPONSE:
Thanks, this will be fixed.

*5) Page 7, line 29: the final sentence makes no sense.*

–AUTHORS' RESPONSE:

[Figure]
* * *
[4]https://www.kwantcontrols.com/product/systems/integrated-telegraph-system/

Thanks for feedback, we will explain this better, as reported below:
First, we recall that VISIR-1.b graph pruning methodology leaves in the graph both sea and land arcs not intersecting the shoreline. At the beginning of the execution of the code for track computation, such a graph is employed for determining, for each of the requested track endpoints (i.e., start and end location of the route), what is its next node on the graph. This can even be a land and not a sea node.

In a subsequent step, the graph arcs are screened for the condition UKC $= z - T > 0$ (Mannarini et al., 2016a, Eq.44). Thus, if the start node was found on land (UKC $\leq 0$), no path outgoing from that node can be computed and VISIR quits with a warning. The coordinate of the requested endpoint has then to be shifted by the VISIR user, in order its next node not to be on land any more.

In an operational use, where the user would set the endpoints for just a single computation, this may be a disturbing feature and will be improved in next VISIR version. For the current assessment exercise, whereby the endpoint are chosen just once and then used for many computations (288 tracks per route, cf. Sect.4.5), we think this approach is still acceptable.

–MANUSCRIPT PARTS INVOLVED:
Sect.2.3, last paragraph.

*6) Page 10, line 14: "anthropic" -> "anthropogenic".*

–AUTHORS' RESPONSE:
Thanks: "climate change of anthropic origin" will be changed into "anthropogenic climate change".

*7) Page 26, line 25: please specify which version of Matlab.*

–AUTHORS' RESPONSE:
Matlab 2016a was used on both the workstation (Mac OS 10.11.6 "El Capitan", employed for the performance analysis of Sect.3.2) and the cluster (Unix CentOS release 6.9 "Final", employed for mass production of Sect.4). In addition, the MEXCDF library is required. Furthermore, the list of all third-party Matlab functions is provided along with the VISIR-1.b release (https://zenodo.org/record/2563074).

−MANUSCRIPT PARTS INVOLVED:
We are going to add this information in the "Code and data availability" section.

*8) Figure 5: the geodetic curves look piecewise linear (i.e. local geodetics between waypoints) rather than continuous - why?*

−AUTHORS' RESPONSE:
Flattening of the geodetic and the piecewise linear geometry of the tracks are due to the finite angular resolution of the graph. In particular, for Fig.5 and 7 a graph with order of connectivity $\nu = 8$ is employed, resulting in an angular resolution $\Delta\theta \sim 7$ (cf. Eq.13).

−MANUSCRIPT PARTS INVOLVED:
We will expand related explanation in Sect.4.3.

*9) Figure 7: the captions refer us to an external website for the animations. I think they should probably refer us to the supplementary material instead.*

−AUTHORS' RESPONSE:
In the caption of Fig.7 we will add a reference to the Supplementary Material. However, we would also like to keep reference to the TIB website which is recommended by Geosci. Model Dev.'s official guidelines for videos[5].

−MANUSCRIPT PARTS INVOLVED:
Caption of Fig.7.
* * *
[5]https://www.geoscientific-model-development.net/for_authors/manuscript_preparation.html

*10) Figure 7: "oncean" -> "ocean" in the ordinate label.*

–AUTHORS' RESPONSE:
Thanks, this will be fixed.

**References**

Bertram, V. and Couser, P.: Computational Methods for Seakeeping and Added Resistance in Waves, in: 13th International Conference on Computer and IT Applications in the Maritime Industries, Redworth, 12-14 May 2014, edited by Volker, B., pp. 8–16, Technische Universität Hamburg- Harburg, 2014.

Fujii, M., Hashimoto, H., and Taniguchi, Y.: Analysis of satellite AIS Data to derive weather judging criteria for voyage route selection, TransNav: International Journal on Marine Navigation and Safety of Sea Transportation, 11, 2017.

Harvald, S. A.: Resistance and propulsion of ships, Krieger Publishing Company, 1992.

IMO: MSC 76/23/Add.1 Resolution MSC.137(76), Annex 6 - Standards for ship manoeuvrability, Tech. rep., International Maritime Organization, London, UK, 2002.

Mannarini, G., Pinardi, N., Coppini, G., Oddo, P., and Iafrati, A.: VISIR-I: small vessels – least-time nautical routes using wave forecasts, Geoscientific Model Development, 9, 1597–1625, https://doi.org/10.5194/gmd-9-1597-2016, http://www.geosci-model-dev.net/9/1597/2016/, 2016a.

Mannarini, G., Turrisi, G., D'Anca, A., Scalas, M., Pinardi, N., Coppini, G., Palermo, F., Carluccio, I., Scuro, M., Cretì, S., Lecci, R., Nassisi, P., and Tedesco, L.: VISIR: technological infrastructure of an operational service for safe and efficient navigation in the Mediterranean Sea, Natural Hazards and Earth System Sciences, 16, 1791–1806, https://doi.org/10.5194/nhess-16-1791-2016, http://www.nat-hazards-earth-syst-sci.net/16/1791/2016/, 2016b.

Mannarini, G., Carelli, L., Zissis, D., Spiliopoulos, G., and Chatzikokolakis, K.: Preliminary inter-comparison of AIS data and optimal ship tracks, TransNav, 13, 53–61, https://doi.org/10.12716/1001.13.01.04, 2019.

Meehl, G. A.: Characteristics of surface current flow inferred from a global ocean current data set, Journal of Physical Oceanography, 12, 538–555, 1982.

Munk, W. H.: Origin and generation of waves, Tech. rep., SCRIPPS INSTITUTION OF OCEANOGRAPHY LA JOLLA CALIF, 1951.

Richardson, P. L.: Drifting in the wind: leeway error in shipdrift data, Deep Sea Research Part I: Oceanographic Research Papers, 44, 1877–1903, 1997.

Techy, L.: Optimal navigation in planar time-varying flow: Zermelo's problem revisited, Intelligent Service Robotics, 4, 271–283, 2011.

---

## Author Comment (AC3) · 20 May 2019

**General assessment**

*The paper is a through one. There are very few papers on ship weather routing covering so many aspects of this optimization problem and doing it with so much detail. I particularly appreciate:*

- *the time interpolation - I agree that it may bring significant benefits for drastically changes in the subsequent weather forecasts,*

- *using bathymetric database with detailed real data,*

- *detailed results and analysis of time savings attributed to exploitation of waves*

[Figure]

*and currents.*

*My specific comments are few – I provide them below.*

−AUTHORS' RESPONSE:
We thank the Referee for his/her time and comments on our manuscript: They definitively contributed to improve it. In this document, we report Referee's text in italics and our replies as a normal text, distinguishing wherever needed our response from the manuscripts parts involved by changes. All references to sections, equations, figures, and tables are relative to the submitted gmd-2018-292 manuscript.

**Specific comments**

**1** - *'We use throughout this manuscript the words "track" or "trajectory" for indicating a set of waypoints joining two given endpoints or harbours, in relation to departure on a given date, and the words "route" or "crossing" when there is no reference to a specific departure date.' While 'track' is perfectly acceptable here, I suggest replacing 'trajectory' with some other word (e.g. path). The word 'trajectory' is usually used in control and robotics with a different meaning: it involves greater accuracy (manoeuvrability and actuation is- sues), especially for obstacle avoidance or collision avoidance purposes. A "trajectory" between two harbours does not make sense.*

−AUTHORS' RESPONSE:
Agreed: occurrences of "trajectory" will be replaced by "path".

–MANUSCRIPT PARTS INVOLVED:
Whole manuscript.

**2** - *Regarding section 2.3: an alternative approach would be to use varying resolution of a graph – the nodes can be placed with larger resolution in coastal areas and with lower resolution at open waters. I suggest commenting on the those two possible approaches to this problem and explaining why you choose the one with additional intersection check.*

–AUTHORS' RESPONSE:
In fact, we took into consideration the fact that the VISIR graph grid may deserve a redesign, e.g. reducing the density of gridpoints in open seas through the use of a nonuniform mesh. An adaptive refinement mesh (Berger and Colella, 1989) or unstructured mesh limiting the minimum angle (Shewchuk, 2002) could be another option. Their advantage would be to reduce the number of open-ocean edges, reducing RAM allocation and speeding up the computation of the shortest path.

However, we point out that, for the safety of navigation, a check on intersection between graph arcs and shoreline is in any case needed, no matter the grid resolution or structure. In fact, even if the mesh is built via a tessellation, intersection with islands and boundary elements smaller than mesh elements should be checked (Legrand et al., 2000). For a graph of higher order of connectivity ($\nu \gg 1$, cf. manuscript's Sect.2.3) this is even more challenging. Such a check on shoreline intersection can easily represent a significant computational cost (De Berg et al., 1997). In order to perform it effectively, it is crucial to be able to find indexes of graph elements next the shoreline. On a regular grid this operation can be carried out in $\mathcal{O}(M)$ time ($M$ is the number of shoreline elements), no matter the size of the maritime domain (and we exploited this in the *i)* step of the algorithm described in Sect.2.3). Instead, on a random or not regular

mesh, a $\mathcal{O}(M \cdot n)$ time would be required by a linear search ($n$ is here either the number of nodes or arcs of the graph). To speed up the search on a not regular mesh, a preliminary node indexing can be computed. With a *k-d* tree, an additional $\mathcal{O}(n \log(n))$ time for tree construction and, on average, $\mathcal{O}(M \cdot \log(n))$ for querying would be needed (Bentley, 1975). This is in excess of the $\mathcal{O}(M)$ estimate for corresponding step (cf. *i*) in Sect.2.3) in present VISIR graph creation algorithm.

Thus, at this stage we still preferred keeping a regular grid which enabled a relatively quick and easy graph computation at the cost of a longer path computing time. This is not critical, given the not operational functioning of VISIR for the present exercise. In future model versions, also depending on coding options, domain, and type of application, we may reconsider this choice.

–MANUSCRIPT PARTS INVOLVED:
Sect.2.3 will be expanded using response above.

**3** - *Regarding section 2.5.2: 'Edges which, for a given EOT, violate stability are pruned before the shortest path algorithm is run. This way, it is ensured that the optimal track preserves vessel intact stability.' Based on the above description, I am not sure if this approach is correct. In presence of coastline, shallows etc. the exact time at which an edge will be transited cannot be know exactly prior to running the algorithm. Even for open ocean, avoiding a cyclone may cause a delay resulting in reaching a certain graph node much later, thus making all prior assumptions inaccurate. Therefore, in my opinion the edges' weights should be verified dynamically during the algorithm run instead of pruning the edges before the run.*

–AUTHORS' RESPONSE:

In VISIR, there is no prior assumption about the vessel time of sailing at the various spatial positions of the domain.

Following (Mannarini et al., 2016, Sect.2.2.2 & pseudocode in App.A), all vessel speeds at any location and direction (i.e. on each of the $A$ edges) and any time ($N_t$ time steps) are computed ahead of path optimization. RAM space allocation for storage of this information is discussed in Sect.3.2, Fig.3 and in our answer to Referee's comment *4a)* below. Then, the time-dependent Dijkstra's algorithm (Mannarini et al., 2016) can manage all this spatially and temporally dependent information for computing the time-optimal paths. Its correctness is demonstrated by comparison with the path resulting from the benchmark solution in a dynamic flow field by Techy (2011) (Sect.3.1.2, Fig.2, Tab.2). Thus, we can say that if cyclone avoidance causes a delay in reaching a specific location, vessel speed at that actually delayed time is used by VISIR for evaluating if sailing through that specific location at that specific time will still be part of the time-dependent optimal path.

For pruning of edges leading to loss of vessel intact stability, the algorithmic machinery works pretty much the same, with specific edges being labeled as unsafe at specific time steps only, cf. (Mannarini et al., 2016, Sect.2.2.2). If a vessel sails at that edge and time, it would experience stability loss, no matter the previous and subsequent path. Thus, that edge is pruned for just that time step ahead of path optimization.

−MANUSCRIPT PARTS INVOLVED:
The description provided in Sect.2.5.2 will be expanded making use of the response above.

**4a** - *While I appreciate the computational complexity analysis based on RAM allocation data, I would also hope for assessing computational time and space based on the algorithm itself. I agree that it is a hard task for complex algorithms, but still some*

*analysis could be made, at least for the worst case.*

–AUTHORS' RESPONSE:
Some deepenings concerning computational (CPU) time and memory space (RAM) of VISIR shortest path algorithm are provided in the following:

- **CPU time**
  Fig.3a (red markers) shows that the worst-case estimate of present VISIR implementation of Dijkstra's time-dependent algorithm scales nearly linearly with the number of degrees of freedom (DOF) of the problem. DOF is proportional to the product of the number $A$ of graph edges and the number $N_t$ of time steps of the dynamic environmental fields. $N_t$ is roughly constant for a given route, as in Fig.3. It can be shown that, upon generalizing the graph arc arrangement of (Mannarini et al., 2016, Fig.1) to any order of connectivity $\nu$ of the graph (cf. Sect.2.3), $A$ is given by

$$A = 4\nu(\nu + 1)N \tag{1}$$

with the number $N$ of graph grid nodes (Mannarini et al., 2018, in review). In any two-dimensional regular mesh, $N$ scales quadratically with the inverse mesh resolution, $N \sim (1/\Delta_g)^2$. For the series of experiments in Fig.3, we varied $\nu$ as $1/\Delta_g$. When taken together, these two effects result into:

$$\text{DOF} = A \cdot N_t \sim \nu^2 N \sim (1/\Delta_g)^4 = \mathcal{O}(N^2) \tag{2}$$

Thus, the empirically retrieved linearity of CPU time with DOF corresponds to a quadratic dependence in $N$. This is in fact the expected worst-case performance of a Dijkstra's algorithm (Bertsekas, 1998). As we stated in Sect 2.4, in presence of binary heaps, such estimate can be reduced to $N \log N$. This will come up in future VISIR versions.

- **RAM allocation**
  In oder to further clarify the memory space requirements of VISIR, with a focus on its shortest path algorithm, we collected and analyzed additional datasets as described below. They consist of:

  $d_1$) time series of RAM allocation of the VISIR Matlab job[1]

  $d_2$) stopwatch timer readings at specific VISIR processing phases[2]

  The $d_2$) dataset is then temporally offset by matching the end of the $d_1$) dataset. Finally, resulting $d_2$) data are smoothed by thinning and this results in the plots displayed in Fig. .e-f below.

  For each graph angular resolution (indexed by $\nu$ parameter) the timeseries exhibit different relative importance (both in terms of duration and RAM allocation) of the various processing phases. However, the $d_1$) and $d_2$) datasets confirm that, for $6 \leq \nu \leq 9$, the peak RAM is allocated during the edge weight computation. Furthermore, the shortest path algorithm is run twice: in its static version (Dijkstra, 1959) for the computation of the geodetic track, in a time-dependent version for the optimal track (Mannarini et al., 2016). The latter requires in input the edge delays at $N_t$ time steps, and this justifies the uphill RAM step between these two phases.

–MANUSCRIPT PARTS INVOLVED:
The information already provided in Sect.3.2 and will be integrated with material above. In particular:
- Fig.3.a-d and Tab.3 will be updated for using performance data from the latest code version and for accounting for smoothing of the RAM timeseries;
* * *
[1]Using the shell command: `top | grep MATLAB >> RAM-timeseries.txt`
[2]Using the Matlab commands: `tic, toc`

- two panels *e)* and *f)* will be added to Fig.3 of the manuscript with following caption:
"a) CPU time for the total VISIR job (blue markers) and for just the computation of the time-dependent shortest path (red markers). Only the *cw* case is shown. Dashed lines are fits of the model in Tab.3. b) Peak RAM allocation during the jobs of a) panel, with a reference line at the total installed RAM. c) Ratio of CPU times of the *cw* to the *w* case and (just for optimal path) for with to without time-interpolation. d) Ratio of peak RAM allocation of the *cw* to *w* type jobs. For panels a,b,d) both cases with (filled) and without (empty markers) time-interpolation. The DOF (Sect.3.2) of the time-dependent shortest path problems is displayed on the horizontal axis. e,f) Time series of RAM memory allocation during VISIR execution for *w* and *cw* type jobs, respectively. Black circles (blue lines) refer to runs without (with) time-interpolation of edge weights. Vertical dashed lines separate the main phases of the processing. Both panels refer to the $\nu = 8$ case of a)-d). The processing phase labels are: *ew* (computation of edge-averaged fields); *ed* (edge delays); *gdt* (geodetic track); *opt* (optimal track)."

**4b** - *It would also be interesting to compare the computational time with that of a non-deterministic approach (there are multiple meta-heuristics available, including Evolutionary Multi-objective Optimization, Ant Colony Optimization etc.).*

−AUTHORS' RESPONSE:
We would like to note first that, being based on Dijkstra's algorithm, VISIR solution is not just guaranteed to be exact, but also its performance (for a given route and vessel departure date) is stable over different runs. This is a difference with evolutionary (EA) and, generally speaking, with heuristics-based algorithms. For that class of algorithms, both the quality and the computational cost of the solution may vary over subsequent runs, as they are driven by random effects. The issue of randomness can be mitigated by statistical averaging over many simulations. However, a more fundamental issue is

that, as clearly stated in Eiben et al. (2003), performance of an EA should be assessed in terms of both efficiency (CPU time) and effectiveness (quality of the solution). Furthermore, even for a specific EA and EA implementation, performance may vary with tuning. Tuning refers to specifying values for the algorithm parameters, such as the "mutation rate". Tuning may affect both EA performance and robustness (Eiben et al., 2003).

Apart from the EA peculiarities, performance comparison of VISIR with other ship routing systems is also hampered by the fact that:

*i)* there is usually little or no evidence that those models were preliminarily validated versus exact solutions;

*ii)* the input environmental fields are not always available for other published results;

*iii)* access to the source code for running on identical conditions would be necessary;

*iv)* the computational platforms employed are either different or not documented;

In a dedicated collaborative effort for evaluation of VISIR vs. a deterministic path planning model which was previously tested against an analytical benchmark, we were able to overcome most of these difficulties (Mannarini et al., 2018, in review). We are open to reply that approach for EA-based ship routing models, e.g., the multi-objective EA reported in (Szlapczynska, 2015) or the ant-colony algorithm described in Tsou and Cheng (2013).

−MANUSCRIPT PARTS INVOLVED:
The information already provided in Sect.3.2 and will be integrated with the discussion above.

**5** - *I agree with the authors that the paper would further benefit from a more realistic modeling of speed loss in waves and wind. I encourage them to include such modelling in their research.*

–AUTHORS' RESPONSE:
Thanks for the comment. In fact such a more realistic modeling of speed loss in waves and wind is planned, at least for Ro-Pax vessels, in the frame of the newly started GUTTA project[3].

–MANUSCRIPT PARTS INVOLVED:
Reference to GUTTA project will be added to the Conclusions.
* * *
[3]http://bit.ly/guttaproject

**Table 3.** Fit parameters for the data displayed in Fig.3a. The fit model is $a \cdot x^b + c$. For the optimal path data, $c$ parameter is not fitted.

|  | units | no T-interp | | with T-interp | |
|---|---|---|---|---|---|
|  |  | optimal path | total job | optimal path | total job |
| $a$ | s | $9.9 \cdot 10^{-8}$ | $4.7 \cdot 10^{-10}$ | $2.6 \cdot 10^{-6}$ | $1.2 \cdot 10^{-7}$ |
| $b$ | – | 1.07 | 1.42 | 1.01 | 1.18 |
| $c$ | s | - | 52 | - | 60 |
| rmse | s | 3.9 | 15.6 | 3.3 | 24.8 |

**References**

Bentley, J. L.: Multidimensional Binary Search Trees Used for Associative Searching, Commun. ACM, 18, 509–517, https://doi.org/10.1145/361002.361007, http://doi.acm.org/10.1145/361002.361007, 1975.

Berger, M. J. and Colella, P.: Local adaptive mesh refinement for shock hydrodynamics, Journal of computational Physics, 82, 64–84, 1989.

Bertsekas, D.: Network Optimization: Continuous and Discrete Models, Athena Scientific, Belmont, Mass. 02178-9998, USA, 1998.

De Berg, M., Van Kreveld, M., Overmars, M., and Schwarzkopf, O.: Computational geometry, in: Computational geometry, pp. 1–17, Springer, 1997.

Dijkstra, E. W.: A note on two problems in connexion with graphs, Numerische mathematik, 1.1, 269–271, 1959.

Eiben, A. E., Smith, J. E., et al.: Introduction to evolutionary computing, vol. 53, Springer, 2003.

Legrand, S., Legat, V., and Deleersnijder, E.: Delaunay mesh generation for an unstructured-grid ocean general circulation model, Ocean Modelling, 2, 17–28, 2000.

Mannarini, G., Pinardi, N., Coppini, G., Oddo, P., and Iafrati, A.: VISIR-I: small vessels – least-time nautical routes using wave forecasts, Geoscientific Model Development,

9, 1597–1625, https://doi.org/10.5194/gmd-9-1597-2016, http://www.geosci-model-dev.net/9/1597/2016/, 2016.

Mannarini, G., Subramani, D., Lermusiaux, P., and Pinardi, N.: Graph-Search and Differential Equations for Time-Optimal Vessel Route Planning in Dynamic Ocean Waves, IEEE Transactions on Intelligent Transportation Systems, 2018, in review.

Shewchuk, J. R.: Delaunay refinement algorithms for triangular mesh generation, Computational geometry, 22, 21–74, 2002.

Szlapczynska, J.: Multi-objective weather routing with customised criteria and constraints, The Journal of Navigation, 68, 338–354, 2015.

Techy, L.: Optimal navigation in planar time-varying flow: Zermelo's problem revisited, Intelligent Service Robotics, 4, 271–283, 2011.

Tsou, M.-C. and Cheng, H.-C.: An Ant Colony Algorithm for efficient ship routing, Polish Maritime Research, 20, 28–38, 2013.

————————————————————

**Fig. 3.** e,f) Time series of RAM memory allocation during VISIR execution for w and cw type jobs, respectively. Black circles (blue lines) refer to runs without (with) time-interpolation of edge weights.

---

## Author Response (AR1)

Geosci. Model Dev. Discuss. **gmd-2018-292**

**VISIR-1.b: ocean surface gravity waves and currents for energy efficient navigation**

G. Mannarini, L. Carelli

Latest revision:
- Place: Lecce, Italy
- Date: 2019-06-03

**Reply to Referee #1**

**General assessment**

*This is a reasonably good paper that describes a new version of a ship-routing model. The original model was published in GMD, so the subject matter has already been judged to fall within the scope of the journal. The manuscript assesses the impact of waves and currents on transatlantic crossings, and calculates energy efficiency savings that seem impressive.*

−AUTHORS' RESPONSE:

We thank the Referee for his/her time and comments on our manuscript: They definitely contributed to improve it. In this document, we report Referee's text in italics and our replies as a normal text, distinguishing wherever needed our response from the manuscripts parts involved by changes. All references to sections, equations, figures, and tables are relative to the submitted gmd-2018-292 manuscript.

**Specific comments**

**1** -*The term "waves" is used throughout the manuscript, but it is never properly defined. The ocean supports a wide variety of wave motions, both internal and at the surface, including gravity waves, Rossby waves, Kelvin waves, Poincare waves, acoustic waves, etc. I believe the manuscript is referring exclusively to surface gravity waves, but this needs to be stated. In the equivalent atmospheric problem of aircraft routing, "waves" usually refers to Rossby waves in the jet stream, and the wave structure (in other words, the $u(x,y,t)$ and $v(x,y,t)$ velocity field associated with the wave) is used in the calculation of the fastest route. I presume that the flow perturbation associated with the surface gravity waves in the current manuscript is not being used like this, but rather that the waves are being treated as areas of turbulence to be avoided. However, this wasn't entirely clear to me and deserves to be clarified.*

−AUTHORS' RESPONSE:

In this manuscript we refer to and use numerical fields of ocean surface gravity waves.

This is in fact the kind of wave motion contributing to so called "wave added resistance" of a vessel. Such resistance is in physical terms a force leading to involuntary vessel speed loss in waves. This force is traditionally distinguished into a radiation (energy dissipated due to vessel heave and pitch motions) and a diffraction component (energy dissipated by the hull to deflect incoming waves, short with respect to vessel length). As in Mannarini et al. (2016a), diffraction is neglected in the present parametrization (Sect.2.5.1). However, for both radiation and diffraction motions, the relevant wave length scale is set by vessel length (which is up to a few hundred m).

Furthermore, in VISIR we do account also for vessel intact stability (Sect.2.5.2), which sets a time scale given by the vessel natural roll period (usually up to about 20 s, or more than 0.05 Hz). The CMEMS wave fields used in VISIR stem from the Météo-France model, which considers a wave spectrum discretized into 24 directions and 30 frequencies in the [0.035 − 0.58] Hz range[1]. Classically, this is the realm of ocean surface gravity waves (Munk, 1951).

−MANUSCRIPT PARTS INVOLVED:

We propose to make changes in following parts:

- Title, to be changed into "VISIR-1.b: ocean surface gravity waves and currents for energy efficient navigation".

- Sect.1.1. Add specification of what ocean waves are considered.
* * *
[1]https://bit.ly/2KWCHYL

– Sect.4.1.4. Add information that the wave spectrum is discretized into 24 directions and 30 frequencies starting from 0.035 Hz to 0.58 Hz[2], comparing to typical vessel natural roll frequencies.

**2** - *The manuscript is missing a discussion on whether the ship-routing model is intended for operational use or just for research purposes. More generally, it is missing a discussion on how ships are currently routed operationally: are the tracks optimal in some sense? If so, who calculates the optimal routes, and using what model? This is particularly relevant to interpret the energy efficiency gains calculated in the manuscript.*

–AUTHORS' RESPONSE:

VISIR can be used either with analysis or forecast environmental fields, since this is not constrained by any of the equations of Sect.2. Thus, VISIR can help for both assessment of past tracks (as in the present work) or prediction of optimal ones (as actually done in the operational system for the Mediterranean Sea described in Mannarini et al. (2016b)).

In the mapping exercise in the Atlantic Ocean included in the present manuscript, for the reason discussed in Sect.4.5. (duration of the transatlantic crossing exceeding maximum lead time of wave forecasts) and the fact that an operational system was not required by the funding project, we resorted to analysis fields. We think this approach can be useful for ex-post assessments of energy efficiency savings. To this end, the main limitation of our approach is the parametrization of speed loss in surface gravity waves (cf. Question 1) above and Sect.2.5.1), which suffers from still large uncertainties in the literature for the wave added resistance (Bertram and Couser, 2014). Energy efficiency gains resulting from VISIR refer to comparison of the least-time to the orthodromic path and their entity also depends on the amount of speed loss in waves.

Concerning the degree of optimization of actually sailed ship tracks, this is an open research question. Weather ship routing systems are used both offshore and onboard for planning, but the final decision is up to the shipmaster (Fujii et al., 2017). Furthermore, route planning may involve sensitive commercial information that a ship operator will not easily share. Thus, the extent to which a ship track is optimized is not always publicly known.

We have recently addressed this question by comparing VISIR optimal tracks vs. reported ship tracks per AIS (Automated Identification System) data, for a route in the Southern Ocean (Mannarini et al., 2019). By computing both spatial and temporal discrepancies between VISIR and AIS tracks, we could infer that optimization likely took place in several but not all tracks. While the method by Mannarini et al. (2019) is still in its infancy, we believe its extension to a larger statistics could contribute to shed light on questions like the one posed in this Referee's comment.

–MANUSCRIPT PARTS INVOLVED:

Sect.4.5

**3** - *I generally found the manuscript difficult to read and understand, mostly because of the poor quality of English usage throughout. This problem could and should be fixed by calling on a native English speaker or professional proof-reading service.*

–AUTHORS' RESPONSE:

Thanks for feedback. We have appointed a professional proof-reading service for reviewing the final version of the manuscript.

**Minor Comments**

*1) Page 1, line 20: "which capacity" -> "whose capacity".*
* * *
[2]https://bit.ly/2KWCHYL

−AUTHORS' RESPONSE:
Thanks, now fixed.

*2) Page 2, line 2: please define "dead reckoning".*
−AUTHORS' RESPONSE:
5 Dead reckoning refers to the computation of a vessel's position by means of establishing its previously known position and advancing it, based on its estimated speed and course over elapsed time. In the study of Richardson (1997), Ship drift (SD) was defined as the difference in the velocity vector between two position fixes and the velocity vector resulting from dead reckoning. In Meehl (1982) a similar definition of SD was given, with the specification that dead reckoning must be computed 24 h in advance of the latest position fix.

10 −MANUSCRIPT PARTS INVOLVED:
Above clarification now added in the Introduction.

*3) Page 4, lines 15-16: what are the manoeuvrability and actuation issues that arise, and what are the consequences of not considering them?*
−AUTHORS' RESPONSE:
15 VISIR computes heading and fraction (EOT) of maximum engine power to be held along an optimal ship track.
In order to head as prescribed by the optimal track, the ship has to be manoeuvred (e.g. acting on rudder and/or lateral thrusters, Bertram (2000)). The rudder is handled via a hydraulic device that converts pressure into a mechanical action to move the rudder[3]. In order to implement the prescribed EOT, the high level order from the control bridge is transmitted through potentiometers[4] to the main engines (and possibly also to other components of the propulsion system such as clutches,
20 gearbox, controllable pitch propeller, cf. Harvald (1992)).
Motions of the bottom layer (rudder, main engine), as related to electro-mechanical devices, should occur on a much shorter timescale (probably seconds to a few minutes) than the top level controls needed for implementing the optimal track (requiring changes in the order of minutes, cf. $ROT_M$ in Tab.7, to hours, cf. Fig.6). Thus, a routing system must ensure that the top level control requires feasible manoeuvers (e.g. in Sect.4.3.2 we check that maximum vessel Rate of Turn $ROT_M$ is in an acceptable
25 range; other feasibility criteria are defined in IMO (2002)). If this condition is satisfied, it should be possible, for the sake of computation of the optimal track, to safely ignore the temporal dynamics of the underlying actuation level (Techy, 2011). On the other hand, if the actuator time scale were comparable to the time over which heading and EOT changes should take place, the hypothesis of top-bottom level separation would be invalid. We presume that this is much less likely to occur in open-sea navigation (which is the subject of the present manuscript) than, for example, during harbour operations. However, on board
30 data would be needed for a thorough assessment of this issue.

−MANUSCRIPT PARTS INVOLVED:
Appendix B added with contents from above discussion.

*4) Page 7, line 17: "preliminary" -> "preliminarily".*
−AUTHORS' RESPONSE:
35 Thanks, now fixed.

*5) Page 7, line 29: the final sentence makes no sense.*
−AUTHORS' RESPONSE:
Thanks for feedback, we will explain this better, as reported below:
First, we recall that VISIR-1.b graph pruning methodology leaves in the graph both sea and land arcs not intersecting the
40 shoreline. At the beginning of the execution of the code for track computation, such a graph is used for determining, for each
* * *
[3]https://www.wartsila.com/encyclopedia/term/rudder-actuator
[4]https://www.kwantcontrols.com/product/systems/integrated-telegraph-system/

of the requested track endpoints (i.e., start and end location of the route), what is its next node on the graph. This can even be a land rather than a sea node.

In a subsequent step, the graph arcs are screened for the condition $UKC = z - T > 0$ (Mannarini et al., 2016a, Eq.44). Thus, if the start node was found on land ($UKC \leq 0$), no path outgoing from that node can be computed and VISIR quits with a
5  warning. The coordinate of the requested endpoint has then to be shifted by the VISIR user, in order its next node not to be on land any more.

In an operational use, where the user would set the endpoints for just a single computation, this may be a disturbing feature and will be improved in next VISIR version. For the current assessment exercise, whereby the endpoint are chosen just once and then used for many computations (288 tracks per route, cf. Sect.4.5), we think this approach is still acceptable.

10  −MANUSCRIPT PARTS INVOLVED:
Sect.2.3.

*6) Page 10, line 14: "anthropic" -> "anthropogenic".*
−AUTHORS' RESPONSE:
Thanks: "climate change of anthropic origin" now changed into "anthropogenic climate change".

15  *7) Page 26, line 25: please specify which version of Matlab.*
−AUTHORS' RESPONSE:
Matlab 2016a was used on both the workstation (Mac OS 10.11.6 "El Capitan", used for the performance analysis of Sect.3.2) and the cluster (Unix CentOS release 6.9 "Final", used for mass production of Sect.4). In addition, the MEXCDF library is required. Furthermore, the list of all third-party Matlab functions is provided along with the VISIR-1.b release (https:
20  //zenodo.org/record/2563074).

−MANUSCRIPT PARTS INVOLVED:
This information now added in the "Code and data availability" section.

*8) Figure 5: the geodetic curves look piecewise linear (i.e. local geodetics between waypoints) rather than continuous - why?*
−AUTHORS' RESPONSE:
25  Flattening of the geodetic and the piecewise linear geometry of the tracks are due to the finite angular resolution of the graph. In particular, for Fig.5 and 7 a graph with order of connectivity $\nu = 8$ is used, resulting in an angular resolution $\Delta\theta \sim 7°$ (cf. Eq.13).

−MANUSCRIPT PARTS INVOLVED:
Related explanation in Sect.4.3 now expanded.

30  *9) Figure 7: the captions refer us to an external website for the animations. I think they should probably refer us to the supplementary material instead.*
−AUTHORS' RESPONSE:
In the caption of Fig.7 we will add a reference to the Supplementary Material. However, we would also like to keep reference to the TIB website which is recommended by Geosci. Model Dev.'s official guidelines for videos[5].

35  Caption of Fig.7.

*10) Figure 7: "oncean" -> "ocean" in the ordinate label.*
−AUTHORS' RESPONSE:
Thanks, now fixed.
* * *
[5]https://www.geoscientific-model-development.net/for_authors/manuscript_preparation.html


**Reply to Referee #2**

**General assessment**

*The paper is a through one. There are very few papers on ship weather routing covering so many aspects of this optimization problem and doing it with so much detail. I particularly appreciate:*

- *the time interpolation - I agree that it may bring significant benefits for drastically changes in the subsequent weather forecasts,*

- *using bathymetric database with detailed real data,*

- *detailed results and analysis of time savings attributed to exploitation of waves and currents.*

*My specific comments are few – I provide them below.*

−AUTHORS' RESPONSE:

We thank the Referee for his/her time and comments on our manuscript: They definitively contributed to improve it. In this document, we report Referee's text in italics and our replies as a normal text, distinguishing wherever needed our response from the manuscripts parts involved by changes. All references to sections, equations, figures, and tables are relative to the

15 submitted gmd-2018-292 manuscript.

**Specific comments**

**1** - *'We use throughout this manuscript the words "track" or "trajectory" for indicating a set of waypoints joining two given endpoints or harbours, in relation to departure on a given date, and the words "route" or "crossing" when there is no reference*

20 *to a specific departure date.' While 'track' is perfectly acceptable here, I suggest replacing 'trajectory' with some other word (e.g. path). The word 'trajectory' is usually used in control and robotics with a different meaning: it involves greater accuracy (manoeuvrability and actuation issues), especially for obstacle avoidance or collision avoidance purposes. A "trajectory" between two harbours does not make sense.*

25 −AUTHORS' RESPONSE:

All occurrences of "trajectory" now replaced by "path".

**2** - *Regarding section 2.3: an alternative approach would be to use varying resolution of a graph – the nodes can be placed with larger resolution in coastal areas and with lower resolution at open waters. I suggest commenting on the those two possible approaches to this problem and explaining why you choose the one with additional intersection check.*

−AUTHORS' RESPONSE:

Following Mannarini et al. (2016), we took into consideration the fact that the VISIR graph grid may need to be redesigned, e.g. by reducing the density of gridpoints in open seas through the use of a nonuniform mesh. An adaptive refinement mesh (Berger and Colella, 1989) or unstructured mesh limiting the minimum angle (Shewchuk, 2002) could be another option. This would

35 reduce the number of open-ocean edges, thereby reducing RAM allocation (cf. Sect.3.2.2) and speeding up the computation of the shortest path.

In any case, to ensure navigation safety, the intersection between graph arcs and shoreline (Sect.2.3) needs to be verified, irrespectively of the grid resolution or structure. In fact, even if the mesh is built via a tessellation, intersection with islands

and boundary elements smaller than mesh elements should be checked (Legrand et al., 2000). For a graph of higher order of connectivity ($\nu \gg 1$) this is even more challenging. Such a check on shoreline intersection can easily represent a significant computational cost (De Berg et al., 1997). In order to perform it effectively, it is crucial to be able to find indexes of graph elements next the shoreline. On a regular grid this operation can be carried out in $\mathcal{O}(M)$ time ($M$ is the number of shoreline elements), irrespectively of the size of the maritime domain (and we exploited this in the *i*) step of the algorithm described in Sect.2.3). Instead, on a random or not regular mesh, a $\mathcal{O}(M \cdot n)$ time would be required by a linear search ($n$ is here either the number of nodes or arcs of the graph). To speed up the search on a not regular mesh, a preliminary node indexing can be computed. With a *k-d* tree, an additional $\mathcal{O}(n \log(n))$ time for tree construction and, on average, $\mathcal{O}(M \cdot \log(n))$ for querying would be needed (Bentley, 1975). This is in excess of the $\mathcal{O}(M)$ estimate for corresponding step (cf. *i*) in Sect.2.3) in the present VISIR graph creation algorithm.

Thus, at this stage we still use a regular grid which enables a relatively quick and easy graph computation at the cost of a longer path computing time. This is not critical, given the non-operational functioning of VISIR for the present exercise. In future model versions, also depending on coding options, domain, and type of application, we may reconsider this choice.

−MANUSCRIPT PARTS INVOLVED:
Appendix C added with contents from above discussion.

**3** - *Regarding section 2.5.2: 'Edges which, for a given EOT, violate stability are pruned before the shortest path algorithm is run. This way, it is ensured that the optimal track preserves vessel intact stability.' Based on the above description, I am not sure if this approach is correct. In presence of coastline, shallows etc. the exact time at which an edge will be transited cannot be know exactly prior to running the algorithm. Even for open ocean, avoiding a cyclone may cause a delay resulting in reaching a certain graph node much later, thus making all prior assumptions inaccurate. Therefore, in my opinion the edges' weights should be verified dynamically during the algorithm run instead of pruning the edges before the run.*

−AUTHORS' RESPONSE:
In VISIR, there is no prior assumption about the vessel time of sailing at the various spatial positions of the domain. Following (Mannarini et al., 2016, Sect.2.2.2 & pseudocode in App.A), all vessel speeds at any location and direction (i.e. on each of the $A$ edges) and any time ($N_t$ time steps) are computed ahead of path optimization. A time-dependent Dijkstra's algorithm (Mannarini et al., 2016) can then manage all this spatially and temporally dependent information for computing the time-optimal paths. Its correctness is demonstrated by comparison with the path resulting from the benchmark solution in a dynamic flow field (Sect.3.1.2, Fig.2, Tab.2).

Similarly, edges that, for a given EOT, violate stability are pruned before the shortest path algorithm is run. Stability loss is assumed to be local in both space and time, no matter what the previous path is before the vessel sails through the edge violating stability. Thus, the edge is pruned only for that time step, ahead of path optimization.

−MANUSCRIPT PARTS INVOLVED:
Sect.2.5.2.

**4a** - *While I appreciate the computational complexity analysis based on RAM allocation data, I would also hope for assessing computational time and space based on the algorithm itself. I agree that it is a hard task for complex algorithms, but still some analysis could be made, at least for the worst case.*

−AUTHORS' RESPONSE:
Some deepenings concerning computational (CPU) time and memory space (RAM) of VISIR shortest path algorithm are provided in the following:

- **CPU time**

  Fig.3a (red markers) shows that the worst-case estimate of present VISIR implementation of Dijkstra's time-dependent algorithm scales nearly linearly with the number of degrees of freedom (DOF) of the problem. DOF is proportional to the product of the number $A$ of graph edges and the number $N_t$ of time steps of the dynamic environmental fields. $N_t$ is roughly constant for a given route, as in Fig.3. It can be shown that, upon generalizing the graph arc arrangement of (Mannarini et al., 2016, Fig.1) to any order of connectivity $\nu$ of the graph (cf. Sect.2.3), $A$ is given by

  $$A = 4\nu(\nu + 1)N \qquad (1)$$

  with the number $N$ of graph grid nodes (Mannarini et al., 2018, in review). In any two-dimensional regular mesh, $N$ scales quadratically with the inverse mesh resolution, $N \sim (1/\Delta_g)^2$. For the series of experiments in Fig.3, we varied $\nu$ as $1/\Delta_g$. When taken together, these two effects result into:

  $$\text{DOF} = A \cdot N_t \sim \nu^2 N \sim (1/\Delta_g)^4 = \mathcal{O}(N^2) \qquad (2)$$

  Thus, the empirically retrieved linearity of CPU time with DOF corresponds to a quadratic dependence in $N$. This is in fact the expected worst-case performance of a Dijkstra's algorithm (Bertsekas, 1998). As we stated in Sect 2.4, in presence of binary heaps, such estimate can be reduced to $N \log N$. This will come up in future VISIR versions.

- **RAM allocation**

  In oder to further clarify the memory space requirements of VISIR, with a focus on its shortest path algorithm, we collected and analyzed additional datasets as described below. They consist of:

  $d_1$) time series of RAM allocation of the VISIR Matlab job[1]

  $d_2$) stopwatch timer readings at specific VISIR processing phases[2]

  The $d_2$) dataset is then temporally offset by matching the end of the $d_1$) dataset. Finally, resulting $d_2$) data are smoothed by thinning and this results in the plots displayed in Fig. 3.e-f below.

  For each graph angular resolution (indexed by $\nu$ parameter) the timeseries exhibit different relative importance (both in terms of duration and RAM allocation) of the various processing phases. However, the $d_1$) and $d_2$) datasets confirm that, for $6 \leq \nu \leq 9$, the peak RAM is allocated during the edge weight computation. Furthermore, the shortest path algorithm is run twice: in its static version (Dijkstra, 1959) for the computation of the geodetic track, in a time-dependent version for the optimal track (Mannarini et al., 2016). The latter requires in input the edge delays at $N_t$ time steps, and this justifies the uphill RAM step between these two phases. Finally, Fig.3.e-f proves that time interpolation does not affect RAM allocation but solely CPU time.

**Table 3.** Fit parameters for the data displayed in Fig.3a. The fit model is $a \cdot x^b + c$. For the optimal path data, $c$ parameter is not fitted.

| | units | no T-interp | | with T-interp | |
| | | optimal path | total job | optimal path | total job |
| --- | --- | --- | --- | --- | --- |
| $a$ | s | $9.9 \cdot 10^{-8}$ | $4.7 \cdot 10^{-10}$ | $2.6 \cdot 10^{-6}$ | $1.2 \cdot 10^{-7}$ |
| $b$ | — | 1.07 | 1.42 | 1.01 | 1.18 |
| $c$ | s | - | 52 | - | 60 |
| rmse | s | 3.9 | 15.6 | 3.3 | 24.8 |

−MANUSCRIPT PARTS INVOLVED:

Sect.3.2, In particular:
* * *
[1]Using the shell command: `top | grep MATLAB >> RAM-timeseries.txt`
[2]Using the Matlab commands: `tic, toc`

[Figure]

**Figure 3.** a) CPU time for the total VISIR job (blue markers) and for just the computation of the time-dependent shortest path (red markers). Only the *cw* case is shown. Dashed lines are fits of the model in Tab.3. b) Peak RAM allocation during the jobs of a) panel, with a reference line at the total installed RAM. c) Ratio of CPU times of the *cw* to the *w* case and (just for optimal path) for with to without time-interpolation. d) Ratio of peak RAM allocation of the *cw* to *w* type jobs. For panels a,b,d) both cases with (filled) and without (empty markers) time-interpolation. The DOF (Sect.3.2) of the time-dependent shortest path problems is displayed on the horizontal axis. e,f) Time series of RAM memory allocation during VISIR execution for *w* and *cw* type jobs, respectively. Black circles (blue lines) refer to runs without (with) time-interpolation of edge weights. Vertical dashed lines separate the main phases of the processing. Both panels refer to the $\nu = 8$ case of a)-d). The processing phase labels are: *ew* (computation of edge-averaged fields); *ed* (edge delays); *gdt* (geodetic track); *opt* (optimal track).

- Fig.3.a-d and Tab.3 will be updated for using performance data from the latest code version and for accounting for smoothing

of the RAM timeseries;

- two panels *e)* and *f)* will be added to Fig.3 of the manuscript as in Fig. 3 above.

**4b** - *It would also be interesting to compare the computational time with that of a non-deterministic approach (there are multiple meta-heuristics available, including Evolutionary Multi-objective Optimization, Ant Colony Optimization etc.).*

−AUTHORS' RESPONSE:

We would like to note first that, being based on Dijkstra's algorithm, VISIR solution is not just guaranteed to be exact, but also its performance (for a given route and vessel departure date) is stable over different runs. This is a difference with evolutionary (EA) and, generally speaking, with heuristics-based algorithms. For that class of algorithms, both the quality and the computa-
10   tional cost of the solution may vary over subsequent runs, as they are driven by random effects. The issue of randomness can be mitigated by statistical averaging over many simulations. However, a more fundamental issue is that, as clearly stated in Eiben et al. (2003), performance of an EA should be assessed in terms of both efficiency (CPU time) and effectiveness (quality of the solution). Furthermore, even for a specific EA and EA implementation, performance may vary with tuning. Tuning refers to specifying values for the algorithm parameters, such as the "mutation rate". Tuning may affect both EA performance and
15   robustness (Eiben et al., 2003).

Apart from the EA peculiarities, performance comparison of VISIR with other ship routing systems is also hampered by the fact that:

   *i)* there is usually little or no evidence that those models were preliminarily validated versus exact solutions;

   *ii)* the input environmental fields are not always available for other published results;

20   *iii)* access to the source code for running on identical conditions would be necessary;

   *iv)* the computational platforms employed are either different or not documented;

In a dedicated collaborative effort for evaluation of VISIR vs. a deterministic path planning model which was previously tested against an analytical benchmark, we were able to overcome most of these difficulties (Mannarini et al., 2018, in review). We are open to reply that approach for EA-based ship routing models, e.g., the multi-objective EA reported in (Szlapczynska,
25   2015) or the ant-colony algorithm described in Tsou and Cheng (2013).

−MANUSCRIPT PARTS INVOLVED:

Appendix D added with contents from above discussion.

**5** - *I agree with the authors that the paper would further benefit from a more realistic modeling of speed loss in waves and wind. I encourage them to include such modelling in their research.*

−AUTHORS' RESPONSE:

Thanks for the comment. In fact such a more realistic modeling of speed loss in waves and wind is planned, at least for Ro-Pax vessels, in the frame of the newly started GUTTA project[3].

−MANUSCRIPT PARTS INVOLVED:

35   Reference to GUTTA project now added to the Conclusions.
* * *
[3]http://bit.ly/guttaproject

**Code and data availability**
p.31, l.1-3: added information on Matlab version and operating systems
p.31, l.6: Updated reference to support assets for the figures and tables on zenodo

**Acknowledgements**
p.31, l.10-11: added mention of two people

**Figures**
p.40: panels a-d) updated with values for latest code version
p.40: added panels e) and f)
p.40: expanded caption for panels e,f)

**Tables**
p.50: values updated following Fig.3 updates
p.52: values updated for referring to actual graph grid point selected by VISIR

**Appendix**
p.50, l.1-: added Section "Note on manoeuvring and actuation"
p.51, l.7: added Section "Note on alternative graph meshes"
p.52, l.11: added Section "Note on model performance comparison"

[revised manuscript text omitted]